# SEESAW: DO GRAPH NEURAL NETWORKS IMPROVE NODE REPRESENTATION LEARNING FOR ALL?

## ABSTRACT

Graph Neural Networks (GNNs) have manifested significant proficiency in various graph learning tasks over recent years. Owing to their exemplary performance, GNNs have garnered increasing attention from both the research community and industrial practitioners. Consequently, there has been a notable transition away from the conventional and prevalent shallow graph embedding methods. However, in tandem with this transition, an imperative question arises: do GNNs always outperform shallow embedding methods in node representation learning? Despite the doubts cast by multiple recent studies, the field of graph machine learning still lacks a systematic understanding, which is essential for meticulously paving its advancement. To properly answer this question, in this work, we propose a principled framework that unifies the pipelines of representative shallow graph embedding methods and GNNs. With rigorous comparative analysis, we first characterize the primary differences in their design from two different perspectives: the prior of node representation learning, and the neighborhood aggregation mechanism. We then analyze the benefits and drawbacks of using GNNs instead of shallow embedding methods through comprehensive experiments on ten real-world graph datasets. Furthermore, we also empirically validate that our analysis can be generalized to GNNs under various learning paradigms. Armed with these insights, we propose a guide for practitioners in choosing appropriate graph representation learning models under different scenarios.

## 1 INTRODUCTION

Graph-structured data is ubiquitous across a plethora of applications, including recommender systems (Ying et al., 2018b; Fan et al., 2019; Sankar et al., 2021; Tang et al., 2022), predictive user behavior models (Pal et al., 2020; Zhao et al., 2021; Tang et al., 2020), and chemistry analysis (You et al., 2018; Li et al., 2018). To gain deeper understanding on graph data and exploit the rich relational information, there has been a surge of interest in learning informative representations for graphs (Hamilton et al., 2017a; Xu et al., 2019). These methods typically learn representations via optimizing mappings that encode nodes or subgraphs as data points in a low-dimensional hidden space (Kipf & Welling, 2017). Their primary goal is to preserve as much task-relevant information of the graph (e.g., the proximity of nodes over the graph topology) as possible. Once the mapping is optimized, the learned representations can serve as the input features to perform a wide spectrum of downstream tasks on graphs, such as node classification (Kipf & Welling, 2017; Hamilton et al., 2017a) and link prediction (Zhang & Chen, 2018; Zhao et al., 2022).

In general, commonly used graph representation learning methods in practice can be divided into two categories, i.e., *shallow graph embedding methods* and *deep graph learning methods* (Hamilton et al., 2017b). Shallow graph embedding methods are mostly characterized by using an embedding lookup table as the mapping from nodes to their representations. For example, DeepWalk (Perozzi et al., 2014) and node2vec (Grover & Leskovec, 2016) directly consider node representations as free parameters. These representations are optimized with a skip-gram model (Mikolov et al., 2013) based on randomly generated walks. On the other hand, deep graph learning methods learn mappings from node attribute space to the latent space. For example, GNNs typically take node attributes and the graph topology as input, and they exploit the topology and attribute information concurrently via neighborhood aggregation. In practice, GNNs are often found to show superior performance in node representation learning to power various tasks over attributed graphs (Dwivedi et al., 2023),

such as node classification (Kipf & Welling, 2017; Xu et al., 2019; Dwivedi et al., 2023), link prediction (Zhang & Chen, 2018; Chamberlain et al., 2023; Zhu et al., 2021b; Ying et al., 2018b), and graph classification (Xu et al., 2019; Ying et al., 2019; You et al., 2021). Such a huge success has made GNNs the most popular graph representation learning methods, attracting increasing attention from researchers and practitioners over recent years (Zhou et al., 2020; Wu et al., 2020).

Nevertheless, close on the heels of the tremendous success of GNNs, several recent studies have revealed that GNNs may also bear worse performance in downstream tasks compared with shallow embedding methods across different scenarios (Wang et al., 2022; Chamberlain et al., 2023; Kipf & Welling, 2016). For example, DeepWalk can easily outperform Variational Graph Auto-Ecoders (Kipf & Welling, 2016), which is commonly believed to exhibit better performance, on multiple real-world graph datasets (Wang et al., 2022). Additionally, multiple other shallow embedding methods (Bordes et al., 2013; Trouillon et al., 2016; Yang et al., 2015; Postavaru et al., 2020) also exhibited superior performances over GNNs in link prediction tasks (Chamberlain et al., 2023). Moreover, graph embedding methods have been widely deployed in various high-stake application scenarios to aid decision making in industry (Dong et al., 2023; Chang et al., 2021). Correspondingly, if practitioners shift from shallow embedding methods to GNNs without careful proof-of-concept evaluations, they could be wasting time and effort as the updated model might result with useless or even erroneous results (Altae-Tran et al., 2017; Chen et al., 2018; Li et al., 2017) if GNNs are not suitable for their data and task. Therefore, given the rising interest in GNNs within the graph machine learning field, there is an urgent need to have a systematic understanding about when GNNs fall short in node representation learning (compared with shallow embedding methods). Although multiple studies have cast doubts on the superiority of GNNs, a systematic study is desired by the community. To bridge this research gap, we ask:

> *When do GNNs exhibit drawbacks compared with shallow embedding methods?*

To answer this question, we pioneer a comprehensive investigation SEESAW (**S**hallow **E**mbedding M**E**thods ve**S**us Gr**A**ph Neural Net**W**orks) to systematically compare the two branches of node representation learning methods. Specifically, we first perform a systematic analysis to compare the pipelines of the two branches with a unified framework. Through such analysis, we attribute the primary differences between shallow embedding methods and GNNs to two factors: *(i)* whether the learning method uses a prior based on node attributes for representation learning; and *(ii)* whether the learning method explicitly performs neighborhood aggregation. Then we present a comprehensive study to compare the performance of methods from the two branches, and explore whether these differences bring drawbacks to GNNs or not. Despite the significant performance superiority of GNNs in most use-cases, we highlight two key drawbacks based on their differences from shallow embedding methods. First, in terms of the learning priors, we found that when only a limited number of attributes are available (i.e., in attribute-poor scenarios), the representations yielded by GNNs usually collapse into a lower-dimensional subspace (instead of spanning the entire available hidden space), a.k.a. dimensional collapse (Zhuo et al., 2023; Jing et al., 2022; He & Ozay, 2022). Second, in terms of neighborhood aggregation, we found that performing aggregation is prone to jeopardizing the performance for certain subgroups, e.g., heterophilic nodes, in downstream tasks.

Armed with the above-mentioned observations, we further present a guide for practitioners to select an appropriate class of representation learning models given their settings. In particular, despite the overall performance superiority of GNNs, we suggest adopting shallow embedding methods instead of more commonly used GNNs in *(i)* attribute-poor scenarios, as shallow embedding methods excel at avoiding dimensional collapse by avoiding using node attributes; *(ii)* highly heterophilic networks, as shallow embedding methods do not perform neighborhood aggregation that jeopardizes the performances of heterophilic nodes.

## 2 PRELIMINARIES

**Notations.** We denote an attributed graph as $\mathcal{G} = \{\mathcal{V}, \mathcal{E}\}$, where $\mathcal{V} = \{v_1, ..., v_n\}$ is the set of $n$ nodes; $\mathcal{E} \subseteq \mathcal{V} \times \mathcal{V}$ is the set of edges. Let $\boldsymbol{A} \in \{0, 1\}^{n \times n}$ and $\boldsymbol{X} \in \mathbb{R}^{n \times c}$ be the adjacency matrix and attribute matrix of $\mathcal{G}$, respectively. Here $n$ represents the total number of nodes, while $c$ is the number of dimensions[1] of the node attributes. In self-supervised node representation learning, an embedding model is denoted as $f_{\boldsymbol{\theta}}$, where $\boldsymbol{\theta}$ denotes the learnable parameters. Specifically, $f_{\boldsymbol{\theta}}$

---

[1]For simplicity, we refer to the total number of dimensions of a space as its dimensionality.

takes a node $v_i$ as input, and outputs its associated embedding. In the node classification task, a decoder typically takes the embedding of a node as input, and outputs the predicted label. In the link prediction task, a decoder typically takes the representations of a pair of nodes as input, and output the predicted probability of being connected.

**Shallow Embedding Methods.** Common shallow embedding methods include those based on matrix factorization and those based on random walks. Without loss of generality, in this paper, we focus on the walk-based ones, since they are observed to yield better performance and thus become the most popular options amongst shallow embedding methods (Hamilton et al., 2017b). Specifically, the mapping from nodes to representations in walk-based shallow embedding methods is usually an embedding lookup table. Such mapping is optimized to extract topological information into node representations. We formulate the mapping as

$$f_{\boldsymbol{\theta}}(v_i) = \boldsymbol{Z}\boldsymbol{v}_i, \tag{1}$$

where $\boldsymbol{Z} \in \mathbb{R}^{d \times n}$ is a matrix of representations, while $\boldsymbol{v}_i \in \mathbb{I}^n$ is a one-hot vector indicating the column in $\boldsymbol{Z}$ associated with node $v_i$. Here the learnable parameter set $\boldsymbol{\theta} = \{\boldsymbol{Z}\}$, which is usually optimized with a walk-based objective. We denote the embedding of node $v_i$ as $\mathbf{z}_{v_i}$ (i.e., the $i$-th column of $\boldsymbol{Z}$). We present a commonly used walk-based objective (Perozzi et al., 2014) as

$$\mathscr{L}_{\text{walk}}\left(\mathbf{z}_{v_i}\right) = -\log\left(\sigma\left(\mathbf{z}_{v_i}^{\top}\mathbf{z}_{v_j}\right)\right) - Q \cdot \mathbb{E}_{v_k \sim P_n(v)}\log\left(\sigma\left(-\mathbf{z}_{v_i}^{\top}\mathbf{z}_{v_k}\right)\right). \tag{2}$$

Here $\mathbf{z}_{v_j}$ denotes the embedding of node $v_j$, which is a node that co-occurs near $v_i$ within a fixed-length random walk; $\sigma(\cdot)$ denotes the activation function; $Q$ is the number of negative samples; $P_n$ is a negative sampling distribution. The topology proximity can be preserved in the embedding of each node via optimizing $\mathscr{L}_{\text{walk}}$ for each node.

**Graph Neural Networks (GNNs).** There have been a plethora of GNNs designed for different purposes over the years. Here we introduce the general paradigm of GNNs (Wu et al., 2020). Typically, a GNN model takes the input $\mathcal{G}$ and outputs $\boldsymbol{Z}$ as the learned embedding matrix for the nodes in $\mathcal{V}$. The basic operation of GNN between $l$-th and $(l+1)$-th layer can be summarized as

$$\boldsymbol{z}_{v_i}^{(l+1)} = \sigma(\text{COMBINE}(\boldsymbol{z}_{v_i}^{(l)}, \text{AGG}(\{\boldsymbol{z}_{v_j}^{(l)} : v_j \in \mathcal{N}(v_i)\}))), \tag{3}$$

where $\boldsymbol{z}_{v_i}^{(l)}$ and $\boldsymbol{z}_{v_i}^{(l+1)}$ is the embedding of node $v_i$ at $l$-th and $(l+1)$-th layer, respectively. In the first layer, $\boldsymbol{z}_{v_i}^{(0)}$ can be initialized as the input node feature $\boldsymbol{x}_{v_i}$. $\mathcal{N}(v_i)$ is the set of one-hop neighbors of $v_i$ according to $\boldsymbol{A}$. AGG$(\cdot)$ represents the aggregating function, e.g., weighted sum. COMBINE$(\cdot)$ is the combining function for output of AGG$(\cdot)$ and $\boldsymbol{z}_{v_i}^{(l)}$, which combines the representation from the centering node and the representations of its neighbors. Various objective functions can be adopted to optimize the learnable parameters of GNNs, including supervised objectives (e.g., cross-entropy loss in classification) and self-supervised ones (e.g., the walk-based objective in Equation (2)).

## 3 SHALLOW EMBEDDING METHODS VS. GNNS: A UNIFIED VIEW

In this section, we design a systematic analysis named SEESAW to characterize the connections and differences between shallow graph embedding methods and GNNs. Based on such analysis, we aim to reveal the benefits and drawbacks of GNNs brought by the identified differences.

To perform rigorous analysis between the two branches of representation learning methods, it is critical to enforce a fair comparison. As discussed in Section 2, most popular shallow embedding methods (e.g., DeepWalk and node2vec) are optimized in a self-supervised learning paradigm with an objective based on random walks On the other hand, GNNs can be optimized either in an end-to-end or self-supervised learning paradigm with various types of objectives. Considering the overlapping in the objectives and the learning paradigms of the two branches, we utilize the widely studied self-supervised learning paradigm with a walk-based optimization objective (Equation (2)) (Hamilton et al., 2017a) to establish a unified view for the purpose of fair comparison.

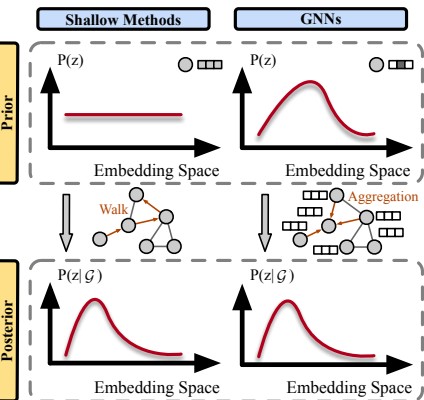

Figure 1: A unified view of pipelines.

We unify the pipelines of the two branches of methods from the perspective of prior-posterior process, with Figure 1 presenting an overview of it. Specif-

ically, we consider the distribution of node representations in the hidden space after model initialization as the prior distribution. For shallow graph embedding methods, as their embedding matrix is randomly initialized, the adopted distribution for node embedding initialization is the prior for representation learning, e.g., a uniform distribution. For GNNs e.g., GCN (Kipf & Welling, 2017) and GraphSAGE (Hamilton et al., 2017a), the node attributes transformed by the randomly initialized learnable parameters are regarded as the prior distribution of node representations in the hidden space. Based on the prior formulations, we characterize the first difference below.

**Difference 1** (Difference in Learning Priors). *Shallow graph embedding methods take any assigned distribution as the prior of representations in the hidden space, while GNNs take the transformed node attributes as the prior of representations.*

Both branches of models perform optimization w.r.t. the graph topology based on the prior. As such, the proximity of nodes over the topology could be preserved in the learned representations. For shallow graph embedding methods, the prior distribution is directly optimized with a walk-based objective function. For GNNs, the output node representations can be optimized with the same objective, but only after the layer-wise neighborhood aggregation is performed. Therefore, we characterize the second difference from the perspective of neighborhood aggregation.

**Difference 2** (Difference in Updating Operations). *Shallow graph embedding methods do not explicitly perform neighborhood aggregation, while GNNs do.*

In the following section, we conduct comprehensive experiments to analyze what the two differences bring to GNNs. Through analysis, we aim to present a systematic understanding of *(i)* the overall superiority of GNNs when node attributes are abundant and *(ii)* the scenarios where shallow embedding methods exhibit superiority while GNNs fall short. Additionally, we note that GNNs can also be optimized with other learning paradigms such as contrastive learning (Zhu et al., 2020b; Ying et al., 2018a) and end-to-end training (Kipf & Welling, 2017). Nonetheless, they generally still adhere to the pipeline delineated in Figure 1. Consequently, we also incorporate them to facilitate a comprehensive and generalizable analysis below.

## 4 EMPIRICAL ANALYSIS: WHAT DO THE DIFFERENCES BRING TO GNNS?

### 4.1 EXPERIMENTAL SETTINGS

**Datasets and Tasks.** We conduct experiments with 10 commonly used real-world benchmark datasets at different scales, including Cora (Yang et al., 2016), CiteSeer (Yang et al., 2016), PubMed (Yang et al., 2016), CoraFull (Bojchevski & Günnemann, 2018), DBLPFull (Bojchevski & Günnemann, 2018), Amazon-Computers (Shchur et al., 2018), Amazon-Photo (Shchur et al., 2018), Coauthor-CS (Shchur et al., 2018), Coauthor-Physics (Shchur et al., 2018), and Flickr (Zeng et al., 2020). More details including dataset statistics are in Appendix B.1. Due to space limit, we only present the most representative results in this section, and we include more comprehensive evaluations in Appendix C.

**Models.** We adopt representative models from shallow embedding methods and GNNs for analysis. Specifically, we utilize DeepWalk (Perozzi et al., 2014) as the representative shallow embedding method, and select GCN with the walk-based loss in Equation (2) (Walk-GCN) as the default GNN for comparison, unless otherwise indicated. To take a step further, we will also present comprehensive empirical results of GNNs in different designs and learning paradigms to demonstrate the generality of our analysis. In terms of GNNs with different designs, there are multiple GNNs designed with contrastive and non-contrastive objectives under the same self-supervised learning paradigm (Shiao et al., 2023). We adopt four representative ones from both branches to study. For contrastive self-supervised GNNs, we adopt GRACE (Zhu et al., 2020b) and a GCN trained with max-margin loss (ML-GCN) (Ying et al., 2018a). For non-contrastive self-supervised GNNs, we adopt Graph Barlow Twins (GBT) (Bielak et al., 2022) and Bootstrapped Graph Latents (BGRL) (Thakoor et al., 2022). In terms of different learning paradigms, we also adopt the vanilla GCN trained in an end-to-end manner (E2E-GCN) for comparison. We report the average results across three separate runs with the corresponding standard deviation. We present more experimental details such as dataset split, evaluation protocol, and implementation details in Appendix B.2.

Table 1: Node classification accuracy comparison between DeepWalk and GCN trained with different learning paradigms on 10 real-world graph datasets, with the best performances highlighted in **Bold**. All numerical numbers of accuracy are in percentages.

|  | Shallow | Walk-GCN | GBT | BGRL | ML-GCN | GRACE | E2E-GCN |
|---|---|---|---|---|---|---|---|
| Cora | $69.33 \pm 0.9$ | $70.07 \pm 1.4$ | $75.63 \pm 1.4$ | $74.33 \pm 1.0$ | $71.87 \pm 1.9$ | $80.87 \pm 0.4$ | $\mathbf{81.40 \pm 0.4}$ |
| CiteSeer | $46.37 \pm 0.9$ | $51.77 \pm 1.4$ | $56.90 \pm 0.8$ | $57.03 \pm 2.8$ | $53.07 \pm 0.3$ | $69.83 \pm 0.7$ | $\mathbf{70.90 \pm 0.5}$ |
| PubMed | $60.77 \pm 0.3$ | $71.13 \pm 2.9$ | $78.63 \pm 0.3$ | $77.27 \pm 3.5$ | $72.03 \pm 1.7$ | $\mathbf{80.50 \pm 0.4}$ | $79.00 \pm 0.4$ |
| CoraFull | $50.37 \pm 1.1$ | $53.77 \pm 0.5$ | $\mathbf{57.69 \pm 0.7}$ | $54.10 \pm 0.9$ | $49.32 \pm 0.3$ | $50.23 \pm 1.1$ | $52.18 \pm 8.3$ |
| DBLPFull | $81.68 \pm 0.7$ | $84.54 \pm 0.4$ | $\mathbf{85.07 \pm 0.0}$ | $84.74 \pm 0.3$ | $81.66 \pm 0.4$ | $83.25 \pm 0.3$ | $85.19 \pm 0.4$ |
| Amz-C. | $88.23 \pm 0.8$ | $88.74 \pm 1.3$ | $89.15 \pm 0.3$ | $88.46 \pm 0.6$ | $89.12 \pm 0.5$ | $86.30 \pm 0.2$ | $\mathbf{91.03 \pm 0.5}$ |
| Amz-P. | $92.57 \pm 0.5$ | $93.64 \pm 0.3$ | $93.07 \pm 0.4$ | $\mathbf{93.68 \pm 0.5}$ | $93.36 \pm 0.5$ | $92.07 \pm 0.2$ | $91.66 \pm 0.6$ |
| Co-CS | $87.69 \pm 0.2$ | $90.08 \pm 0.3$ | $\mathbf{93.75 \pm 0.2}$ | $92.92 \pm 0.1$ | $92.95 \pm 0.2$ | $92.68 \pm 0.6$ | $93.23 \pm 0.1$ |
| Co-Phy. | $93.40 \pm 0.4$ | $95.83 \pm 0.3$ | $95.84 \pm 0.2$ | $95.74 \pm 0.0$ | $95.19 \pm 0.0$ | OOM | $\mathbf{95.86 \pm 0.2}$ |
| Flickr | $\mathbf{52.45 \pm 0.1}$ | $46.26 \pm 0.2$ | $51.87 \pm 0.1$ | $51.89 \pm 0.2$ | $51.16 \pm 0.3$ | OOM | $48.19 \pm 0.2$ |

## 4.2 Overall Performance Evaluation

We first present an overall performance comparison between shallow embedding methods and GNNs in Table 1. Without loss of generality, we take node classification as an exemplary downstream task. We observe that GNNs exhibit significant superiority over shallow embedding methods in most datasets. We assume that such superiority comes from two perspectives, where each perspective associates with one difference identified in Section 3. First, in terms of the learning prior, GNNs are able to exploit the information encoded in the node attributes, which could lead to more task-relevant information in the learned node representations. As a comparison, shallow embedding methods typically are not capable of incorporating node attributes. Second, in terms of neighborhood aggregation, GNNs are able to exploit more abundant localized information by explicitly performing neighborhood aggregation between a node and its direct neighbors. On the other hand, shallow embedding methods preserve the topological proximity only through optimizing the walk-based loss, which may leave relatively more neighbors unexplored. We note that consistent observations have also been reported in recent studies on other graph learning tasks such as link prediction (Shiao et al., 2023; Zhu et al., 2020b; Thakoor et al., 2022). We present more discussions in Appendix C. Interestingly, the only dataset which shallow embedding method exhibits superiority on is Flickr. We attribute such phenomenon to the limited number of available node attributes and less homophily in this dataset, which will be further discussed in the following subsections.

Despite the significant superiority of GNNs over shallow embedding methods, we found GNNs could also exhibit clear drawbacks associated with the two differences. We discuss the drawbacks of GNNs brought by Difference 1 and Difference 2 in the following two subsections.

## 4.3 Are There Any Drawbacks of Using An Attribute-Based Prior?

We note that using an attribute-based prior could enable GNNs to exploit the information from both attributes and graph topology, which usually brings advantages. However, we found that their differences may also jeopardize the performance of GNNs in certain scenarios. Here we focus on the potential drawbacks brought by using a prior based on node attributes in GNNs, i.e., Difference 1. Our rationale is that if the prior is directly obtained based on node attributes, then the performance of GNNs could also heavily rely on the quality of attributes. Nevertheless, we note that in practice, the rich high-dimensional node attributes are not always available. In those attribute-poor scenarios (i.e., when the attributes are only partially available), GNNs may struggle to learn high quality node representations, and thus could also end up with limited performance in downstream tasks. On the contrary, shallow graph embedding methods typically do not bear such a problem since they usually do not rely on node attributes. In light of this, here we study the drawbacks of GNNs in attribute-poor scenarios. We present our observations below, with additional results in Appendix C.

**Observation 1: GNN Performance Drops Under Limited Input Attributes.** We first compare the performance of GNNs under different attribute dimensionalities by manually controlling the number of available attribute dimensions. Specifically, we refer to the ratio of available attribute dimensions for GNN as *attribute dimensionality ratio*, and we vary such a ratio in {100%, 1%, 0.01%} with a minimum number of node attribute dimensionality being one. We present the performance comparison between different ratios of attribute dimensions on node classification and link prediction in Figure 2(a) and Figure 2(b), respectively. We observe that, when using 100% node attributes,

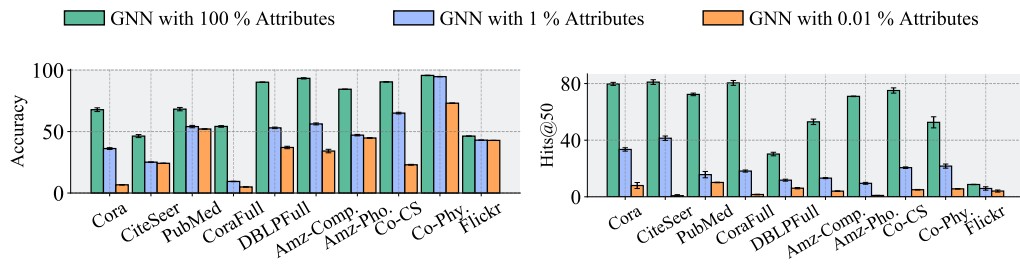

(a) Accuracy in Node Classification.    (b) Hits@50 in Link Prediction.

Figure 2: A comparison between GNNs with different ratios of available node attribute dimensionality in node classification and link prediction. (a): Comparison of node classification accuracy; (b): Comparison of Hits@50 in link prediction.

GNN yields satisfying performance on all datasets, demonstrating the superiority of GNNs when abundant attributes are available. However, when the attribute dimensionality becomes limited, i.e.,

in cases with 1% and 0.01% node attributes, the performance of GNN drops significantly in both tasks. Such a phenomenon demonstrates that limited attribute dimensionality typically jeopardizes the performance of GNNs. As a comparison, shallow embedding methods typically do not bear such an issue, since they usually do not take node attributes as input. In fact, when the attribute dimensionality is limited, the representations yielded by GNNs collapse into a lower-dimensional subspace instead of spanning the entire hidden space, a.k.a. dimensional collapse (Zhuo et al., 2023; Jing et al., 2022; He & Ozay, 2022). We present an illustration in Figure 3, and we elaborate on details in the observation below.

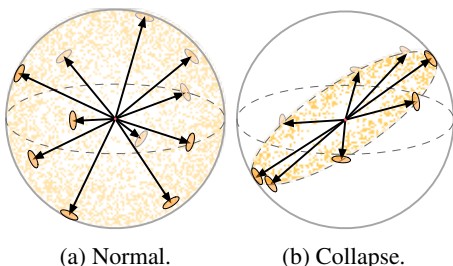

(a) Normal.    (b) Collapse.

Figure 3: An illustration of (a) normal representations in the hidden space vs. (b) representations after dimensional collapse.

**Observation 2: Limited Input Attributes Typically Cause Dimensional Collapse.** To explore to what extent dimensional collapse happens, here we characterize the *effective dimension* as the rank of the learned embedding matrix. Then, we measure the level of dimensional collapse with the ratio of the effective dimension number to the total number of hidden dimensions, i.e., $r/d$, where $r$ is the rank of the node representation matrix and $d$ is dimension of the hidden space. As such, the lower the ratio, the more severe the dimension collapse becomes. Figure 4 presents the effective dimension ratio across different datasets and attribute dimensionalities in node classification task.

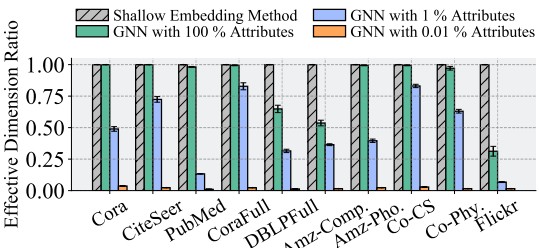

Figure 4: Representation effective dimension ratios.

We observe that more available node attribute dimensions play a critical role in learning representations that span a higher dimensional space, which prevents GNNs from dimensional collapse. As a comparison, shallow embedding method consistently yields a maximal effective dimension ratio, which attributes to the difference in their learning priors. Specifically, shallow embedding methods randomly initialize the representations, and thus it consistently tends to learn full-rank node representation matrix. As such, in attribute-poor scenarios, it's difficult for GNNs to learn representations spanning a higher dimensional space. We present further discussions on the ranks of the learned representations in Appendix C.4.

**Observation 3: Dimensional Collapse Ties to Performance & Attribute Dimensionality.** To further understand the influence of the dimensionality of the representation subspace, we experiment with enforcing a bound over the effective dimensionality of the learned representations. In this way, we are able to manually control the level of dimensional collapse in the learned embedding matrix. Specifically, we propose to consider the learned embedding matrix from GNNs as $\boldsymbol{Z} = \boldsymbol{C}\boldsymbol{F}$, where $\boldsymbol{Z} \in \mathbb{R}^{n \times d}$, $\boldsymbol{C} \in \mathbb{R}^{n \times r}$, and $\boldsymbol{F} \in \mathbb{R}^{r \times d}$ ($1 \leq r \leq d$). As such, $r$ naturally serves as an upper bound of the rank for $\boldsymbol{Z}$ without changing the dimensionality of the hidden space. In practice, we consider the output matrix of GNN model and a matrix with learnable parameters as $\boldsymbol{C}$ and $\boldsymbol{F}$, respectively.

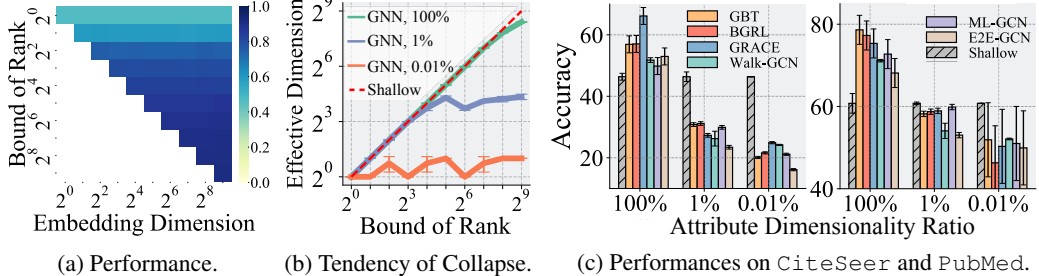

(a) Performance.  (b) Tendency of Collapse.  (c) Performances on `CiteSeer` and `PubMed`.

Figure 5: (a) and (b): Tendencies of performance and collapse under different rank bounds (enforced for embedding matrix). (c): Performance comparison of other popular GNNs across different available attribute dimensionality ratios.

First, we present the tendency of node classification accuracy on `Amazon-Computers` in Figure 5(a), where both $r$ and $d$ vary within a wide range between $2^0$ and $2^9$ to cover most commonly used values. We observe that the performance (indicated by the color) improves as long as the bound of the rank $r$ improves under any embedding dimension number $d$. This further demonstrates that spanning a larger subspace (within the available hidden space) is beneficial for the quality of the learned representations. Second, we present the tendency of effective dimension w.r.t. the bound of rank on `DBLPFull` under different input attribute dimensionalities in Figure 5(b). We observe that, in attribute-poor scenarios, it is difficult for GNNs to learn representations with large effective dimensions even if the bound raises to a larger value. We note that such observations are consistent across datasets and provide additional results and discussion in Appendix C.3 and Appendix C.4.

Finally, Figure 5(c) shows the performances of DeepWalk and GNNs under different learning paradigms under different node attribute availabilities. We can observe that the performance reduction due to dimensional collapse (as discussed above) can also be observed with other GNNs. By contrast, shallow embedding method learns representations spanning a maximum subspace (i.e., effective dimensionality equals to the bound of rank). These observations demonstrate that small attribute dimensionality indeed prevents a wide spectrum of GNNs from learning representations that span a larger subspace, while shallow embedding methods typically do not bear such an issue.

## 4.4 ARE THERE ANY DRAWBACKS OF PERFORMING NEIGHBORHOOD AGGREGATION?

In this subsection, we explore the potential drawbacks brought by performing neighborhood aggregation in GNNs, i.e., Difference 2. We note that explicitly performing neighborhood aggregation enables GNNs to extract information from its direct neighbors. In this way, GNNs allows each node to take advantage of abundant localized information around it. However, multiple existing works have pointed out that the neighborhood aggregation mechanism could jeopardize the performance of GNNs (Zhu et al., 2021a; Luan et al., 2022). For example, when most labels of a node's neighbor are different from its own (i.e., heterophilic cases), the learned embedding associated with this node could be misled by the information aggregated from its neighbors. To exclude the influence of differences in their priors, we propose to adopt *(i)* shallow embedding methods w/ neighborhood aggregation and *(ii)* GNNs w/o neighborhood aggregation for comparison with normal shallow embedding methods and GNNs, respectively. Specifically, for the former, we add a layer of the mean aggregator (Hamilton et al., 2017a) in DeepWalk during both training and inferencing. For the latter, we remove the neighborhood aggregation before the non-linear transformation in each GNN layer.

**Observation 4: Heterophilic Nodes Barely Benefit From Aggregation.** We empirically validate whether neighborhood aggregation mechanism could jeopardize the performance of GNNs or not in the self-supervised learning paradigm. Figure 6(a) and Figure 6(b) present comparisons between (i) shallow embedding methods w/ and w/o neighborhood aggregation and (ii) GNNs w/ and w/o neighborhood aggregation, respectively. The homophily score of each node is measured with the ratio of its neighbors with the same labels (as this node) to the total number of its neighbors (Zhu et al., 2020a). We can observe that the overall performance (i.e., the performance at homophily score equals to one) are similar between w/ and w/o aggregation for shallow embedding method, while the overall performance for GNN w/o aggregation reduces significantly compared with vanilla GNN. This demonstrates that neighborhood aggregation could be critical to effectively exploit the informa-

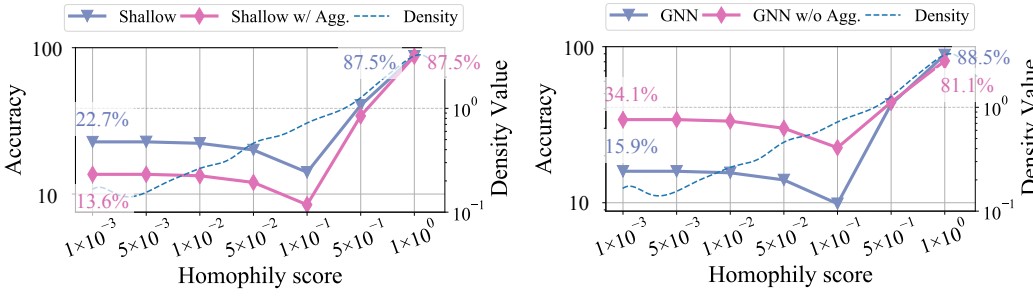

(a) Shallow Methods w/ & w/o Aggregation.  (b) GNNs w/ & w/o Aggregation.

Figure 6: Cumulative node classification accuracy comparison across nodes with different levels of homophily: (a) shallow embedding methods w/ vs. w/o neighborhood aggregation; (b) GNNs w/ vs. w/o neighborhood aggregation. Performance is on `Amazon-Computers` dataset for both figures. The density function for the homophily score of nodes is marked out with the dashed curve.

tion encoded in node attributes. Furthermore, we observe that neighborhood aggregation reduces the performance on nodes with low levels of homophily (i.e., high levels of heterophily) for both models.

Such observation validates that the phenomenon of heterophily nodes suffering from neighborhood aggregation. Similar observations can be found on other datasets, and more results are presented in Appendix C.5.

Finally, Figure 7 shows the performances of different GNNs w/ and w/o neighborhood aggregation for top 10% most heterophilic and homophilic nodes in the `Amazon-Photo` dataset. We can observe that the performance downgrade brought by neighborhood aggregation also widely exists in different GNNs across different learning paradigms.

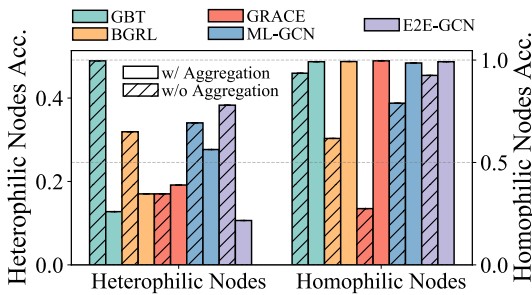

Figure 7: Performance comparison of other GNNs across heterophilic and homophilic nodes.

## 5 DISCUSSION: A GUIDE FOR PRACTITIONERS

Based on the discussion above, we conclude that it is necessary to meticulously select the branches of models to use, instead of adopting GNNs as a panacea. Armed with such insights, in this section, we provide a guide for practitioners to choose an appropriate type of models to learn high-quality node representations, such that the performance in various downstream tasks can be improved.

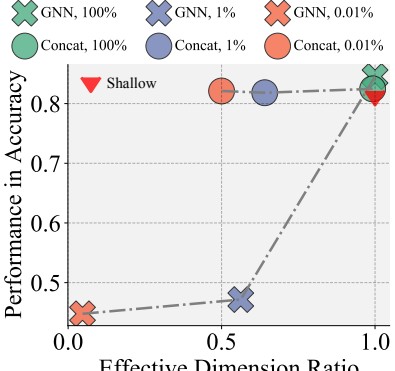

Figure 8: Comparison between GNNs, GNNs with concatenated representations from shallow embedding method (denoted as *Concat*), and shallow embedding methods (denoted as *Shallow*).

It's worth noting that neither shallow embedding methods nor GNNs are flawless: it is difficult for shallow embedding methods to properly exploit information encoded in node attributes, while GNNs also bear drawbacks as discussed in Section 4.3 and Section 4.4. To properly handle their drawbacks, a straightforward way to leverage the advantages from both is by simply combining the representations learned from both types of models. The rationale is two-fold: *(i)* The number of effective dimensions of the learned representations (i.e., the rank of the node representation matrix) from GNNs could be promoted by representations from shallow methods, which helps to tackle the problem of dimensional collapse even in attribute-poor scenarios (i.e., when node attributes are only partially available). *(ii)* Information from both aggregated and unaggregated node attributes could be preserved at the same time, which helps to alleviate the performance reduction on heterophilic nodes.

As an example, we found that simply concatenating the representations from both models helps achieve satisfying performance across different attribute dimensionality ratios. We present a perfor-

mance comparison between shallow method, GNN, and the strategy of using the concatenation of node representations yielded by the two methods on `DBLPFull` dataset in Figure 8. We observe that the performance and effective dimension ratio of GNNs reduce significantly when attribute dimensionality ratio declines. However, by simply combining the representations yielded by the two methods, we are able to significantly reduce its sensitivity to attribute dimensionality ratio, achieving a more stable performance along both axes. We also have similar observations on other datasets. Nevertheless, such combination would result in significantly higher computational complexity as well as the need of maintaining two models instead of one, which is often much less preferred in industrial applications given cost and human resourcing concerns. Hence, it's of practical significance to provide guidance in choosing between the two branches of methods. Below, we formulate the guidance from two perspectives: *(i)*. data perspective, as discussed in previous sections, and *(ii)*. model perspective, which has been extensively discussed in literature.

**Data - Attribute-Rich vs. Attribute-Poor Networks.** As discussed in Section 4.3, GNNs often achieve superior performance in scenarios with rich attribute compared to shallow embedding methods. Correspondingly, adopting GNNs for representation learning on attribute-rich networks is an obvious choice. Nevertheless, in attribute-poor scenarios, e.g., when the attribute dimensionality is limited, GNNs are prone to exhibit dimensional collapse, while shallow embedding methods do not bear such a drawback. Therefore, we recommend adopting GNNs and shallow embedding methods on attribute-rich and attribute-poor networks, respectively.

**Data - Homophilic vs. Heterophilic Networks.** According to discussion in Section 4.3, shallow embedding methods and GNNs exhibit different performance on nodes with different levels of heterophily. In particular, GNNs exhibit superior and inferior performance on homophilic and heterophilic nodes (compared with shallow embedding methods), respectively. A preliminary reason is that explicitly performing neighborhood aggregation is helpful for homophilic nodes while harmful to heterophilic nodes. Therefore, GNNs are recommended for representation learning if the network data is homophilic, otherwise shallow embedding methods could be more suitable.

**Model - Transductive vs. Inductive Settings.** As the shallow embedding methods rely on training an embedding vector for each of the node in the graph, they naturally do not support inductive learning. That is, given any newly appeared nodes, shallow embedding methods cannot produce it's representation without retraining or at least fine-tuning the model. On the other hand, as feature-based models, GNNs are naturally inductive and are able to inference node representations for the newly appeared nodes (Hamilton et al., 2017a). Hence, for use-cases such that the graphs are rapidly updating (e.g., social networks, e-commerce networks, etc), GNNs are recommended given their inductive bias, whereas shallow embedding methods require frequent costly retrains.

**Model - Low-parameter vs. High-parameter Settings.** Modern machine learning usually requires loading all learnable parameters into limited GPU memory to achieve higher training speed. The number of learnable parameters for shallow embedding methods grows linearly with the number of nodes. On the other hand, the parameters size of GNNs is only proportional to the dimension of node attributes and not the number of nodes. Therefore, for large graphs with high number of nodes, GPU training might not be feasible for shallow embedding methods without techniques such as model parallelism (Li, 2023).

## 6 CONCLUSION

In this study, we aim to provide a broader perspective on graph learning, challenging the prevailing emphasis on GNNs. Specifically, we proposed a principled framework that unifies the pipelines of representative shallow graph embedding methods and GNNs. Based on the framework, we preformed systematic comparison between shallow graph embedding methods and GNNs. In essence, we characterized their primary differences from two different perspectives, and analyzed the benefits and drawbacks the two differences bring to GNNs through comprehensive experiments. Notably, the drawbacks are found generalizable onto different GNNs under different learning paradigms, highlighting the practical significance. Armed with these insights, we further discuss a structured guide for practitioners on selecting appropriate graph representation learning models. With this paper, our primary endeavor is to recalibrate the academic perspective, accentuating both the benefits and drawbacks of GNNs compared with conventional shallow embedding methods. We hope our work enlighten practitioners and researchers to foster the meticulous advancement of this field.

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

## A   RELATED WORK

**Shallow Embedding Methods.** Common shallow embedding methods simply consider the mapping from nodes to representations in the hidden space as lookup tables, and the representations are directly optimized w.r.t. the objective function (Hamilton et al., 2017b; Perozzi et al., 2014; Grover & Leskovec, 2016). In general, shallow embedding methods can be divided into two types, i.e., those based on matrix factorization (Ahmed et al., 2013; Belkin & Niyogi, 2001; Cao et al., 2015; Ou et al., 2016) and those based on random walks. Among them, the approaches based on random walks, e.g., DeepWalk (Perozzi et al., 2014) and node2vec (Grover & Leskovec, 2016), have shown superior performance in a plethora of settings.

**Graph Neural Networks.** Graph Neural Networks (GNNs) have emerged to be powerful frameworks to tackle learning problems on graphs (Wu et al., 2020; Zhou et al., 2020; Hamilton et al., 2017a; Kipf & Welling, 2017). GNNs learn low-dimensional representations by extracting information from both attributes and graph topology (Velickovic et al., 2018). Such success can be attributed to its carefully designed neighborhood aggregation, through which a node iteratively extracts information from its direct neighbors (Velickovic et al., 2018; Wu et al., 2020; Zhou et al., 2020). Correspondingly, GNNs have been widely used in many real-world applications (Ying et al., 2018b; Fan et al., 2019; Pal et al., 2020; Zhao et al., 2021; You et al., 2018; Li et al., 2018).

**Comparison of Shallow Embedding Methods vs. GNNs.** Several existing works have compared traditional shallow embedding methods and GNNs from different perspectives, including optimization objectives (Cai et al., 2018; Zhang et al., 2018; Cui et al., 2018; Amara et al., 2021; Hamilton et al., 2017b), properties learned (Cai et al., 2018; Goyal & Ferrara, 2018; Zhang et al., 2018), computational complexity (Goyal & Ferrara, 2018; Cui et al., 2018), and applications (Cai et al., 2018; Amara et al., 2021). Nevertheless, the performance comparison over real-world datasets and practical settings of data availability and heterogeneity are ignored. A few recent studies have performed performance comparison between the two branches (Makarov et al., 2021), with the conclusion that GNNs consistently achieve superior performance. However, we argue they do not sufficiently explore the drawbacks of GNNs. Different from the works above, we present a systematic study to compare the two branches based on both theoretical and experimental discussion. This allows us to *(i)* elaborate on the drawbacks of GNNs in a finer granularity; and *(ii)* propose simple yet effective strategies to tackle the drawbacks.

## B   EXPERIMENTAL SETTINGS

### B.1   DATASETS

We perform the empirical evaluations on 10 real-world network datasets, which span different fields such as scientific publications (citation networks and co-authorship networks) and e-commerce (merchandise networks), including `Cora` (Yang et al., 2016), `CiteSeer` (Yang et al., 2016), `PubMed` (Yang et al., 2016), `CoraFull` (Bojchevski & Günnemann, 2018), `DBLPFull` (Bojchevski & Günnemann, 2018), `Amazon-Computers` (Shchur et al., 2018), `Amazon-Photo` (Shchur et al., 2018), `Coauthor-CS` (Shchur et al., 2018), `Coauthor-Physics` (Shchur et al., 2018), and `Flickr` (Zeng et al., 2020). We present their statistics in Table 2. We directly utilize the APIs provided by PyTorch Geometric (Fey & Lenssen, 2019) to load all datasets.

Among these datasets, `Cora`, `CiteSeer`, `PubMed`, `CoraFull`, and `DBLPFull` are citation networks, where nodes represent documents and edges are citation links. `Amazon-Computers` and `Amazon-Photo` are networks of merchandise, where nodes denote goods and edges represent that two goods are frequently bought together. `Coauthor-CS` and `Coauthor-Physics` are coauthor networks. Here nodes are authors, and two authors are connected if they have co-authored a paper. `Flickr` is a social network of images (as nodes), and a pair of images are connected if they share similar properties such as geographic locations.

### B.2   IMPLEMENTATION DETAILS

**Dataset Split.** We first introduce the dataset split for node classification tasks. For all datasets with available public splits (i.e., `Cora`, `CiteSeer`, and `PubMed`), we utilize the

given public splits to train an MLP model based on the learned node representations and measure the utility such as node classification accuracy. For those datasets without available public splits (i.e., `CoraFull`, `DBLPFull`, `Amazon-Computers`, `Amazon-Photo`, `Coauthor-CS`, `Coauthor-Physics`, and `Flickr`), we utilize a commonly used with 60%/20%/20% split for the train/validation/test split, following the same settings explored by other literature. (Maekawa et al., 2022; Chien et al., 2021) We first introduce the dataset split for link prediction tasks. For all datasets, we explore the random split of 85%/5%/10%, following the same standard as existing works (Shiao et al., 2023; Zhang & Chen, 2018; Cai et al., 2021). Only training edges are visible during the training phase.

**Evaluation Protocol.** We consider the representation learning models as the encoder, and we follow a standard evaluation protocol to train a decoder to perform downstream tasks on graphs based on the learned representations. Specifically, we train an MLP model as the decoder in both node classification and link prediction tasks, which is a commonly adopted approaches in a series of related works (Kipf & Welling, 2016; Tan et al., 2023). In node classification, we input each node embedding into an MLP model and predict the associated label. In link prediction, we take the Hadamard product for representations of each node pair. An MLP takes the resulted vector as input, and output the associated predicted probability of being connected. All results are presented as an average value across three different runs together with the associated standard deviation.

**Details of GNNs for Comparison.** In this paper, we adopt vanilla GCN and vanilla GraphSAGE as the most representative GNNs for comparison. In addition, we also adopted state-of-the-art contrastive and non-contrastive self-supervised learning GNNs. We present a more detailed discussion below. For contrastive self-supervised GNNs, we adopt GRACE (Zhu et al., 2020b) and a GCN trained with max-margin loss, i.e., ML-GCN (Ying et al., 2018a). Specifically, GRACE first generates two correlated graph views by randomly performing corruption. Then, the embedding model is trained with a contrastive loss to maximize the agreement between node representations in these two views. On the other hand, ML-GCN is trained with a walk-based max-margin loss, which forces the agreement between nodes appear in same walks (measured with inner product) to exceed a certain positive margin. For non-contrastive self-supervised GNNs, we adopt Graph Barlow Twins (GBT) (Bielak et al., 2022) and Bootstrapped Graph Latents (BGRL) (Thakoor et al., 2022). Specifically, GBT computes the representations cross-correlation matrix of two distorted views of a single graph. The objective function is formulated to force the cross-correlation matrix to be as close as possible to the identity matrix. In this way, no negative sample is needed for optimization, which improves the practical efficiency. On the other hand, BGRL maintains two distinct graph encoders, and learns the node representations by training an online encoder to predict the embedding of a target node. This also enables BGRL to avoid using negative samples during learning.

**Downstream Tasks and Metrics.** In this work, we use the two most commonly studied tasks for graph data: node classification and link prediction. Following the literature, we use node classification accuracy and F1 score for node classification (Dwivedi et al., 2023), and Hits@50 is adopted for link prediction (Shiao et al., 2023).

**Machine Details.** We ran our experiments on Google Cloud Platform. For all experimental results reported in this paper, we run the corresponding experiments on either NVIDIA P100 or V100 GPUs. Specifically, the machine is configured with 12 virtual CPU cores and 64 GB of RAM for most experiments.

**Open-Sourced Code.** Our open-sourced code is released here: `https://github.com/anonymoussibmissionpurpose/anonymous`.

## C ADDITIONAL EXPERIMENTAL RESULTS AND ANALYSIS

In this section, we present additional experimental results and corresponding analysis. Specifically, we first discuss the overall performance across different GNNs. Then, we present an additional analysis on the GNN performance on nodes with different levels of degrees between attribute-rich and attribute-poor scenarios. After that, we present comprehensive results from both shallow embedding methods and GNNs to demonstrate the relationship between dimensional collapse, embedding dimensionality, and performance. In the last two sections, we first present the relationship between the bound of the rank and the actual rank of the learned node embedding matrix between shallow embedding methods and GNNs. We then discuss the performance difference between (1) shallow

Table 2: Statistics of the 10 real-world datasets we adopted in this paper. We utilize `Amz-C.`, `Amz-P.`, `Co-CS`, and `Co-Phy.` to represent `Amazon-Computers`, `Amazon-Photo`, `Coauthor-CS`, and `Coauthor-Phy`, respectively.

|  | Cora | CiteSeer | PubMed | CoraFull | DBLPFull | Amz-C. | Amz-P. | Co-CS | Co-Phy. | Flickr |
|---|---|---|---|---|---|---|---|---|---|---|
| **#nodes** | 2,708 | 3,327 | 19,717 | 19,793 | 17,716 | 13,752 | 7,650 | 18,333 | 34,493 | 89,250 |
| **#edges** | 10,556 | 9,104 | 88,648 | 126,842 | 105,734 | 491,722 | 238,162 | 163,788 | 495,924 | 899,756 |
| **#features** | 1,433 | 3,703 | 500 | 8,710 | 1,639 | 767 | 745 | 6,805 | 8,415 | 500 |
| **#classes** | 7 | 6 | 3 | 70 | 4 | 10 | 8 | 15 | 5 | 7 |

Table 3: F1 score (macro) comparison between shallow embedding method, vanilla GCN, and other state-of-the-art self-supervised learning GNNs on 10 real-world graph datasets. We highlight the best performances in **Bold**. All numerical numbers of accuracy are in percentages.

|  | **Shallow** | **Walk-GCN** | **GBT** | **BGRL** | **ML-GCN** | **GRACE** | **E2E-GCN** |
|---|---|---|---|---|---|---|---|
| Cora | $69.45 \pm 0.9$ | $69.80 \pm 0.7$ | $74.85 \pm 1.2$ | $74.31 \pm 1.5$ | $67.93 \pm 1.5$ | $79.91 \pm 0.2$ | $\mathbf{80.34 \pm 0.7}$ |
| CiteSeer | $44.82 \pm 1.0$ | $50.21 \pm 1.2$ | $54.65 \pm 0.4$ | $54.38 \pm 2.7$ | $48.39 \pm 1.3$ | $65.51 \pm 1.2$ | $\mathbf{67.30 \pm 0.8}$ |
| PubMed | $58.92 \pm 0.3$ | $70.19 \pm 3.0$ | $77.79 \pm 0.4$ | $76.81 \pm 3.2$ | $71.98 \pm 3.3$ | $\mathbf{81.45 \pm 0.5}$ | $78.49 \pm 0.1$ |
| CoraFull | $29.38 \pm 0.5$ | $32.37 \pm 0.4$ | $39.54 \pm 1.0$ | $37.37 \pm 1.0$ | $33.35 \pm 1.3$ | $32.15 \pm 0.8$ | $\mathbf{40.55 \pm 7.5}$ |
| DBLPFull | $76.26 \pm 0.9$ | $80.32 \pm 0.1$ | $80.44 \pm 0.1$ | $\mathbf{81.22 \pm 0.3}$ | $76.70 \pm 0.6$ | $80.89 \pm 0.1$ | $81.19 \pm 0.3$ |
| Amz-C. | $86.78 \pm 0.9$ | $86.54 \pm 1.4$ | $87.21 \pm 0.7$ | $85.73 \pm 1.3$ | $85.98 \pm 1.2$ | $79.92 \pm 0.3$ | $\mathbf{89.45 \pm 0.4}$ |
| Amz-P. | $91.77 \pm 1.0$ | $\mathbf{92.49 \pm 0.5}$ | $91.30 \pm 0.7$ | $92.28 \pm 0.9$ | $90.45 \pm 0.6$ | $90.31 \pm 0.1$ | $88.60 \pm 0.8$ |
| Co-CS | $84.25 \pm 0.7$ | $86.94 \pm 0.7$ | $91.68 \pm 0.4$ | $90.97 \pm 0.4$ | $89.13 \pm 0.5$ | $\mathbf{91.94 \pm 0.2}$ | $91.40 \pm 0.4$ |
| Co-Phy. | $91.04 \pm 0.5$ | $94.42 \pm 0.3$ | $94.41 \pm 0.3$ | $94.25 \pm 0.0$ | $93.57 \pm 0.3$ | OOM | $\mathbf{94.48 \pm 0.3}$ |
| Flickr | $\mathbf{23.29 \pm 0.0}$ | $16.58 \pm 0.0$ | $21.00 \pm 0.2$ | $21.13 \pm 0.4$ | $20.23 \pm 0.3$ | OOM | $16.41 \pm 0.1$ |

embedding method and that with the enforced neighborhood aggregation; (2) GNNs and that without the neighborhood aggregation.

## C.1 ANALYSIS: PERFORMANCE OF DIFFERENT GNNS

We first present the performance of different state-of-the-art GNNs from a different perspective to reveal their superiority over shallow embedding methods in terms of overall performance. Without loss of generality, we take node classification as an example, and we measure the performance with F1 score (macro). We present the corresponding performance in Table 3. We observe that GNNs exhibit significant performance superiority over shallow embedding method on almost all datasets, which remains consistent with the discussion in Section 4.2. It is worth note that the only dataset where shallow embedding method exhibits superiority over all other GNNs is `Flickr`, where we have the smallest available node attribute dimensionality (see Table 2). This implies that such superiority could be undermined when only limited node attribute dimensionality is available, which is in align with the discussion in Difference 1.

## C.2 ANALYSIS: PERFORMANCE ON NODES WITH DIFFERENT LEVELS OF DEGREES

To gain a deeper understanding of the performance on a finer granularity, we propose to also explore the influence of limited attributes on nodes with different levels of degrees, which allows us to gain an understanding of performance w.r.t. available attribute dimensionality at a fine-grained level.

Specifically, we propose to first compute the ranking of all nodes based on their degree. Then we divide the nodes in the test set into high- and low-degree nodes with a percentile threshold. Without loss of generality, here we take the percentile threshold as 50%. We present the experimental results in Table 4. We observe that low-degree nodes bear more significant accuracy reduction in seven out of the 10 adopted datasets. A potential reason is that low-degree nodes rely more on the information contained in the attribute-based prior (than high-degree nodes). Specifically, compared with high-degree nodes, low-degree nodes tend to receive relatively less information from its neighbors through the neighborhood aggregation mechanism. Therefore, the information encoded in their attributes could dominate the performance. We have consistent observations in link prediction.

Table 4: Node classification accuracy of GNNs under different numbers of attribute dimensions. We highlight the performances with the most significant reduction when transitioning from using 100% to 1% attributes in **Bold**. All numerical numbers of accuracy are in percentages.

| | GNN + 100% Attributes | | GNN + 1% Attributes | |
|---|---|---|---|---|
| | **High-Degree** | **Low-Degree** | **High-Degree** | **Low-Degree** |
| Cora | $74.7 \pm 1.3$ | $61.0 \pm 1.7$ | $\mathbf{35.6 \pm 0.8 \ (- 52.3 \ \%)}$ | $36.9 \pm 0.7 \ (- 39.6 \ \%)$ |
| CiteSeer | $53.4 \pm 0.9$ | $39.3 \pm 1.6$ | $32.1 \pm 0.4 \ (- 39.9 \ \%)$ | $\mathbf{18.1 \pm 0.1 \ (- 54.1 \ \%)}$ |
| PubMed | $73.1 \pm 0.9$ | $63.5 \pm 2.1$ | $\mathbf{51.9 \pm 1.3 \ (- 29.0 \ \%)}$ | $56.3 \pm 0.6 \ (- 11.3 \ \%)$ |
| CoraFull | $59.3 \pm 0.5$ | $49.1 \pm 1.1$ | $11.8 \pm 0.3 \ (- 80.1 \ \%)$ | $\mathbf{7.09 \pm 0.2 \ (- 85.6 \ \%)}$ |
| DBLPFull | $87.1 \pm 0.3$ | $81.8 \pm 0.2$ | $61.7 \pm 0.8 \ (- 29.2 \ \%)$ | $\mathbf{32.5 \pm 1.4 \ (- 60.2 \ \%)}$ |
| Amz-C. | $92.4 \pm 0.5$ | $88.0 \pm 0.4$ | $63.9 \pm 1.3 \ (- 30.9 \ \%)$ | $\mathbf{42.2 \pm 2.3 \ (- 52.0 \ \%)}$ |
| Amz-P. | $95.6 \pm 0.2$ | $91.0 \pm 0.9$ | $67.1 \pm 1.7 \ (- 29.9 \ \%)$ | $\mathbf{45.4 \pm 0.9 \ (- 50.1 \ \%)}$ |
| Co-CS | $93.0 \pm 0.4$ | $87.9 \pm 0.4$ | $72.7 \pm 1.5 \ (- 21.8 \ \%)$ | $\mathbf{57.3 \pm 0.5 \ (- 34.8 \ \%)}$ |
| Co-Phy. | $97.5 \pm 0.3$ | $93.8 \pm 0.2$ | $96.9 \pm 0.1 \ (- 0.69 \ \%)$ | $\mathbf{92.6 \pm 0.3 \ (- 1.35 \ \%)}$ |
| Flickr | $43.0 \pm 0.2$ | $49.6 \pm 0.1$ | $\mathbf{37.8 \pm 0.1 \ (- 12.1 \ \%)}$ | $48.2 \pm 0.0 \ (- 2.82 \ \%)$ |

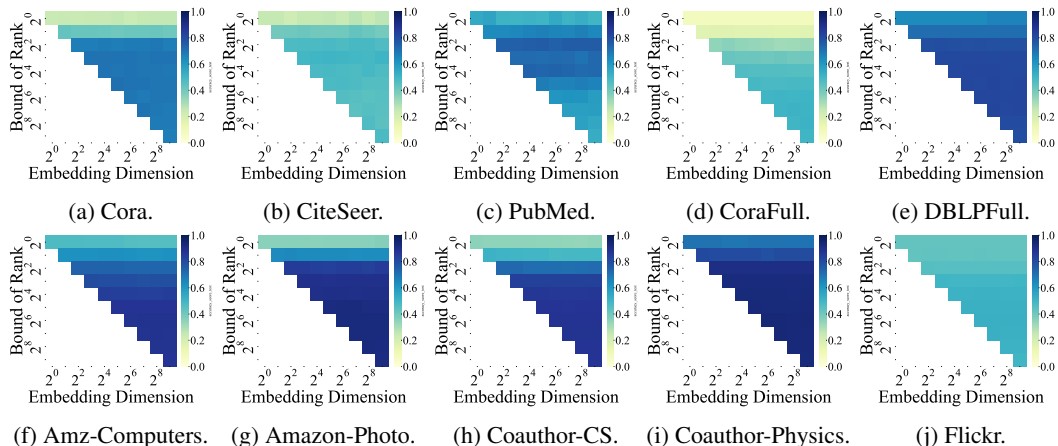

Figure 9: The node classification performance (measured with classification accuracy) of shallow embedding method on 10 different real-world graph datasets. A lower value of *Bound of Rank* implies a heavier level of dimensional collapse. We utilize Amz-Computers to refer to the dataset `Amazon-Computers`.

### C.3 ANALYSIS: DIMENSIONAL COLLAPSE VS. PERFORMANCE

We then present the experimental results of performance w.r.t. the level of dimensional collapse. We utilize the same strategy introduced in *Observation 3* in Section 4.3 to enforce different levels of dimensional collapse. Specifically, instead of directly obtaining the learned node embedding matrix from the GNN model, we consider the learned embedding matrix as $\boldsymbol{Z} = \boldsymbol{CF}$, where $\boldsymbol{Z} \in \mathbb{R}^{n \times d}$, $\boldsymbol{C} \in \mathbb{R}^{n \times r}$, and $\boldsymbol{F} \in \mathbb{R}^{r \times d}$ ($1 \leq r \leq d$). In practice, $\boldsymbol{C}$ is the direct output of the GNN model, while $\boldsymbol{F}$ is a matrix with learnable parameters. We optimize both the learnable parameters in the GNN model and all elements in $\boldsymbol{F}$ during the end-to-end learning process. Without loss of generality, we present two sets of results as representative performances of shallow embedding methods and GNNs: (1) for shallow embedding methods, we present the results from DeepWalk in node classification task as an example in Figure 9.; (2) for GNNs, we present the results from GCN in node classification task as an example in Figure 10. We also have similar observations in link prediction task, other shallow embedding methods, and GNNs. We have the following observations.

First, we observe that in all cases, improving the value of embedding dimension does not significantly change the performance, which implies that the value of embedding dimension does not play a key role in learning high-quality node representations.

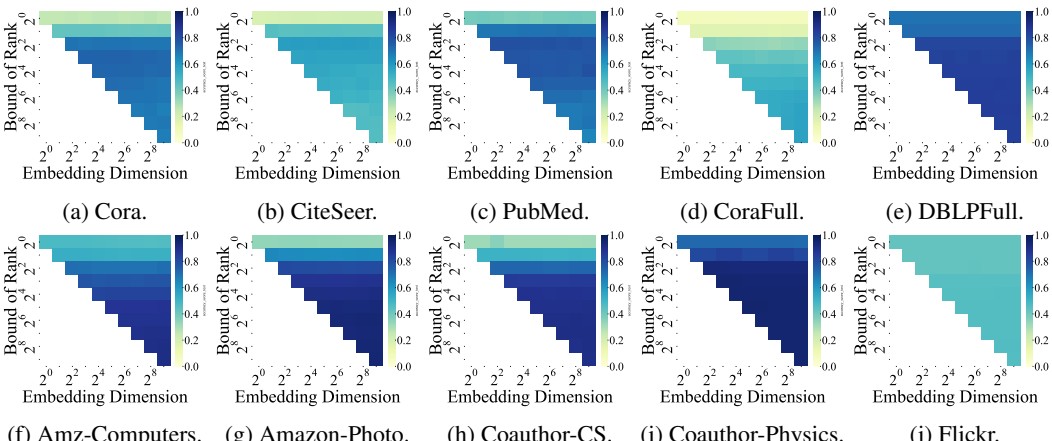

Figure 10: The node classification performance (measured with classification accuracy) of GNN on 10 different real-world graph datasets. Here the available ratio of the node attribute dimensionality is 100%. A lower value of *Bound of Rank* implies a heavier level of dimensional collapse. We utilize Amz-Computers to refer to the dataset `Amazon-Computers`.

Second, we found that in almost all cases, improving the bound of the rank is able to significantly improve the performance (i.e., lead to deeper colors in the heatmap). Here, a smaller value of the bound of the rank generally implies a heavier level of dimensional collapse. This because no matter what value the embedding dimension takes, the learned representations will be guaranteed to collapse into a lower dimensional subspace as long as the the bound of the rank is small. The dimensionality of the lower dimensional subspace is upper-bounded by the bound of the rank. Such an observation demonstrates that as long as dimensional collapse happens, the performance then significantly drops no matter how large the embedding dimensionality is. On the contrary, if dimensional collapse is relieved, the performance is then also improved under a given embedding dimension. We note that improving the bound of the rank could also bring performance reduction in very few cases, e.g., on the PubMed dataset for both shallow embedding methods and GNNs. Such a phenomenon could be caused by overfitting, which goes beyond the scope of this paper.

To summarize, we conclude that no matter what value embedding dimension takes, dimensional collapse always leads to significantly performance drop, which implies a worse quality of the node representations. At the same time, mitigating the dimensional collapse will be beneficial to the performance in most cases. This remains consistent with our conclusion discussed in Section 4.3.

### C.4    ANALYSIS: RANKS OF LEARNED REPRESENTATIONS

We now present an analysis on the rank of the learned node representations between shallow embedding methods and GNNs with different levels of attribute availability. Specifically, we utilize the strategy introduced in Section 4.3 and Appendix C.3 to control the level of dimensional collapse. At each value of the bound of the rank, we calculate the actual rank of the learned node embedding matrix. Note that the actual rank of the learned node embedding matrix cannot exceed the bound of the rank. Correspondingly, in an ideal case, an representation learning model should yield an embedding matrix with a rank equivalent to the bound of the rank, such that the node representations will span a hidden subspace as large as possible. In Figure 11, we present the curves of effective dimension vs. bound of rank for each of the 10 real-world graph datasets. In each subfigure, we present curves from four different scenarios, namely using shallow embedding method, using GNN with 100% attributes, using GNN with 1% attributes, and using GNN with 0.01% attributes. Here, without loss of generality, we adopt DeepWalk and GCN as the representative model for shallow embedding methods and GNNs, respectively. We have the following observations.

First, the curve of effective dimension of shallow embedding method is a straight line of $y = x$ in all cases. This reveals the clear advantage of shallow embedding method in defending against dimensional collapse, since it can always learn node embedding matrices spanning the whole available

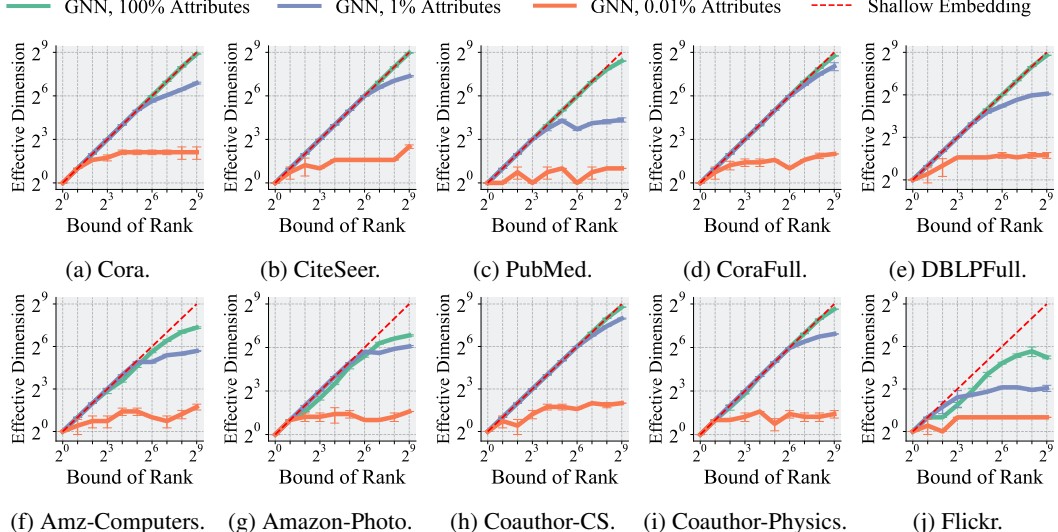

Figure 11: A comparison between shallow embedding method and GNNs under different ratios of available attributes on 10 real-world graph datasets. Here, *Effective Dimension* represents the rank of the learned node embedding matrix, which is upper bounded by the *Bound of Rank*. The dimensionality of hidden space is $2^9$. We utilize Amz-Computers to refer to the dataset `Amazon-Computers`.

hidden subspace. As discussed in Section 4.3, such an advantage can be attributed to avoiding using a prior based on the node attributes.

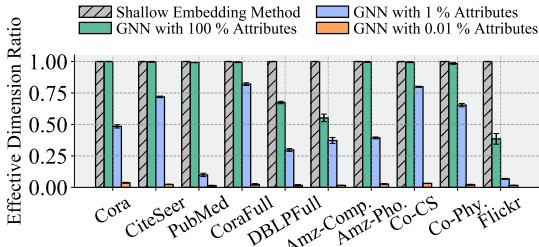

Figure 12: Effective dimension ratio of representations learned for link prediction task.

Second, we found that it is always difficult for GNNs to avoid dimensional collapse when the dimensionality of the input node attributes is limited. We take Cora dataset (given in Figure 11(a)) as an example. GNNs with 100% available node attributes can always achieve a value of effective dimension (i.e., the rank of the learned node embedding matrix) equal to the bound of the rank. Nevertheless, when only 1% node attributes are available, dimensional collapse begins to happen when the bound of rank is larger than $2^6$. When only 0.01% node attributes are available, dimensional collapse even begins to happen when the bound of rank is larger than $2^2$. As a comparison, shallow embedding method can always achieve an effective dimension that is equals to the bound of the rank. This demonstrates that available node attribute dimensionality directly influences the level of dimensional collapse. We also present the effective dimension ratio of the learned node representations in Figure 12, revealing that dimensional collapse also happens under representations learned for link prediction in GNNs, while shallow embedding methods do not encounter such a problem. As discussed in Section 4.3, such a drawback should be attributed to using a prior based on the node attributes. Such a conclusion remains consistent with the discussion in Section 4.3.

## C.5 Analysis: Performance w/ Aggregation vs. Performance w/o Aggregation

We finally present a comparison of the performance between representation learning models with and without neighborhood aggregation. Specifically, we adopt the same strategy as discussed in Section 4.4, where shallow embedding method with neighborhood aggregation (with a mean aggregator) and GNNs without neighborhood aggregation are implemented for comparison. In this way, we are able to rigorously compare the effect of performing neighborhood aggregation within each branch of models, and whether using a prior based on node attributes or not will not influence the conclu-

sion. Without loss of generality, we take DeepWalk and GraphSAGE as the representative example of shallow embedding methods and GNNs, respectively. For any node, we measure the homophily score with the ratio of the total number of its direct neighbors with the same label as itself to the total number of its direct neighbors. We have the following observations.

First, the shallow embedding methods without neighborhood aggregation can achieve better performance on those nodes with small homophily scores in almost all cases. As a comparison, the cumulative performance of shallow embedding methods with neighborhood aggregation improves faster than that of shallow embedding methods without neighborhood aggregation. This indicates a better performance of shallow embedding method with neighborhood aggregation on those nodes with large homophily scores.

Second, we found that a similar phenomenon also exists in GNNs. Specifically, the GNNs without neighborhood aggregation can achieve better performance on those nodes with small homophily scores in almost all cases. As a comparison, GNNs with neighborhood aggregation can always achieve a similar or even better performance compared with that without neighborhood aggregation. This indicates a better performance of GNNs with neighborhood aggregation on those nodes with large homophily scores.

To summarize, we conclude that for both types of models, performing neighborhood aggregation typically helps the performance on those nodes with larger homophily scores, while this could also do harm to the performance on those nodes with smaller homophily scores. This remains consistent with our conclusion discussed in Section 4.4.

## C.6 ANALYSIS: COMBINING SHALLOW EMBEDDING METHODS AND GNNS

We now present the analysis on the performance of combining shallow embedding methods and GNNs by directly concatenating their node representations. The rationale here is two-fold: *(i)* The number of effective dimensions of the learned representations (i.e., the rank of the node embedding matrix) from GNNs could be promoted by representations from shallow methods, which helps to tackle the problem of dimensional collapse in attribute-poor scenarios (mentioned in Section 4.3). *(ii)* Information from both aggregated and unaggregated node attributes could be preserved at the same time, which helps to alleviate the performance reduction on heterophilic nodes (mentioned in Section 4.4). This remains consistent with the disucssion in Section 5. Without loss of generality, we take DeepWalk and GCN as the shallow embedding model and the GNN model, respectively. We take the performance of node classification accuracy on the `DBLPFull` dataset as an example, and present the results across different levels of available attribute dimensionalities in Figure 8. We present the observations below.

First, the performance of GNNs with 100% available node attributes is superior to that of shallow embedding method, which reveals that useful information could be encoded in the node attributes, which contribute to the performance of GNNs.

Second, when the available attribute dimensionality is decreased, the performance of GNNs reduces significantly together with the performance (measured with node classification accuracy). This generally reflects that less available node attributes will typically lead to dimensional collapse, which remains consistent with the discussion in Section 4.3.

Third, when we use the concatenation of the node representations from the two methods, we observe that the performance does not significantly reduce when the available node attributes become limited. This validate the superiority of combining the representations from the two methods across scenarios with different available node attribute dimensionalities.

To summarize, we observe that although GNNs with 100% node attributes can achieve the best performance, its performance reduces significantly once the available node attributes are limited. However, by simply concatenating their representations, we can obtain much more stable performance across scenarios with different available node attribute dimensionalities. However, this approach will lead to a higher computational complexity, since both models need to be optimized. Hence such method can hardly be recommended in industrial settings due to the high computational cost.

Table 5: Performance comparison between shallow embedding method and GNNs on the Squirrel dataset under different levels of attribute availibility ratio.

| | 100%Att, Acc | 100%Att, EDR | 1%Att, Acc | 1%Att, EDR | 0.01%Att, Acc | 0.01%Att, EDR |
|---|---|---|---|---|---|---|
| **GNN** | **38.8%** | 74.2% | 26.3% | 21.9% | 19.0% | 1.56% |
| **Shallow** | 31.5% | **99.6%** | **31.5%** | **99.6%** | **31.5%** | **99.6%** |

Table 6: Node classification accuracy comparison between GNN (w/ Aggregation) and GNN w/o Aggregation on the Squirrel dataset under different levels of node heterophily score.

| | 1e-3 | 5e-3 | 1e-2 | 5e-2 | 1e-1 | 5e-1 | 1e0 |
|---|---|---|---|---|---|---|---|
| **GNN (w/ Aggregation)** | 26.88% | 26.88% | 26.88% | 26.73% | 25.44% | **36.64%** | **38.75%** |
| **GNN w/o Aggregation** | **30.10%** | **30.10%** | **30.10%** | **30.69%** | **26.48%** | 32.20% | 33.17% |

## C.7 SELECTION OF SHALLOW METHOD

We select DeepWalk as a representative shallow graph embedding method to compare with in this paper. The reason why DeepWalk is adopted is that DeepWalk is a representative example of walk-based shallow methods in its design. Specifically, DeepWalk is among the most commonly used shallow graph embedding methods, and a large amount of following works under the umbrella of shallow methods are developed based on DeepWalk. Therefore, DeepWalk is among the best options we can choose to obtain generalizable analysis, and adopting more follow-up methods that share similar design with DeepWalk does not change the observation and conclusion.

## C.8 EXPERIMENTAL RESULTS WITH HETEROPHILIC GRAPHS

We would like to note that our analysis does not depend on whether the adopted datasets are homophilic or not. Here we present the comparison between GNNs and shallow methods on the heterophilic dataset. Specifically, we select the Squirrel dataset and present the corresponding performances below as a representative example, since the Squirrel dataset has a comparable scale (5,201 nodes) with the datasets adopted in our paper and is also highly heterophilic (homophilic ratio 0.22). First, we perform experiments to evaluate dimensional collapse. Here utility is measured by node classification accuracy, while the effective dimension ratio (EDR) is measured by the ratio of the value of rank (of representation matrix) to the representation dimensionality. We present the experimental results in Table 5. We observe a similar tendency as presented in our paper: the GNN model bears severe dimensional collapse when available attributes become limited, while the shallow method is not influenced since it does not take any node attribute as its input.

Second, we also perform experiments to study how the performance changes from highly heterophilic nodes to highly homophilic nodes. We first present the study between GNN (w/ neighborhood aggregation) and GNN w/o neighborhood aggregation. We present their cumulative performances in node classification accuracy under different values of the homophilic score below (the same setting as in Figure 6 in our paper). We present the experimental results in Table 6. We ob-

Table 7: Node classification accuracy comparison between shallow method (w/o Aggregation) and shallow method w/ Aggregation on Squirrel dataset under different levels of node heterophily score.

| | 1e-3 | 5e-3 | 1e-2 | 5e-2 | 1e-1 | 5e-1 | 1e0 |
|---|---|---|---|---|---|---|---|
| **Shallow w/ Aggregation** | 21.86% | 21.86% | 21.86% | 23.23% | 26.35% | **30.36%** | **31.54%** |
| **Shallow (w/o Aggregation)** | **25.14%** | **25.14%** | **25.14%** | **26.26%** | **26.71%** | 29.95% | 31.44% |

Table 8: Node classification accuracy comparison between GNN and GNN under two types of attribute augmentation approaches (including concatenating a random matrix denoted as R and concatenating a matrix of structural features denoted as S onto the original node attribute matrix) on Cora dataset. Here we adopt the walk-based GCN (denoted as W-GCN) as the backbone GNN.

|  | 100%Att, Acc | 100%Att, EDR | 1%Att, Acc | 1%Att, EDR | 0.01%Att, Acc | 0.01%Att, EDR |
|---|---|---|---|---|---|---|
| **W-GCN** | 67.8% | 96.9% | 36.9% | 35.5% | 32.3% | 1.56% |
| **W-GCN (R)** | 68.0% | 97.3% | 29.9% | 67.2% | 10.6% | 5.08% |
| **W-GCN (S)** | 69.0% | 97.3% | 37.3% | 35.2% | 32.3% | 2.34% |

serve a similar tendency as presented in our paper: neighborhood aggregation will jeopardize the performances over those highly heterophilic nodes while benefiting highly homophilic nodes. In addition, we also perform experiments to compare shallow method w/ neighborhood aggregation vs. shallow method w/o neighborhood aggregation. We present the experimental results in Table 7, and the observations remain consistent.

In conclusion, we also have similar observations on heterophilic datasets from both studied perspectives, and our analysis does not depend on whether the adopted datasets are homophilic or not.

### C.9 EXPERIMENTAL RESULTS WITH ATTRIBUTE AUGMENTATION

We performed experiments by (1) concatenating a random matrix with the same dimensionality as the original node attributes onto the node attribute matrix and (2) concatenating a matrix encoded with structural information following the state-of-the-art *degree* strategy (Cui et al., 2022) onto the node attribute matrix. We present the unsupervised learning performances on the Cora dataset below as an example. Here utility is measured by node classification accuracy, while the effective dimension ratio (EDR) is measured by the ratio of the value of rank (of representation matrix) to the representation dimensionality. We present the experimental results in Table 8. We observe that: (1) concatenating a matrix with structural information slightly improves the node classification accuracy, while such a strategy does not stop the significant drop in the rank of node representations; (2) concatenating random node attributes successfully improves the rank of the node representations, however, the classification accuracy is reduced. Therefore, both strategy does not really solve the problem of dimensional collapse, and we believe handling such a problem is non-trivial. Correspondingly, this paper is particularly interesting to researchers working in this area and such a problem is also worth to be explored in future works.

### C.10 OBSERVATIONS OVER OTHER TYPES OF GNNS

**Performance of APPNP.** We perform empirical experiments based on APPNP. We present the unsupervised learning performances on the Cora dataset below as an example. Here utility is measured by node classification accuracy, while the effective dimension ratio (EDR) is measured by the ratio of the value of rank (of representation matrix) to the representation dimensionality. We present the experimental results in Table 9. We observe that similar to GCN, APPNP also bears severe dimensional collapse (exhibited by the significant reduction in the value of EDR).

**Performance of LINKX.** We perform empirical experiments based on LINKX. We present the unsupervised learning performances on the Cora dataset below as an example. Here utility is measured by node classification accuracy, while the effective dimension ratio (EDR) is measured by the ratio of the value of rank (of representation matrix) to the representation dimensionality. We observe that compared with GCN, LINKX exhibits smaller values of EDR in attribute-rich scenarios (e.g., 100% available node attributes), while it also mitigates dimensional collapse in attribute-poor scenarios (e.g., 0.01% available node attributes). This demonstrates that (1) such an approach may jeopardize the effective dimension ratio in attribute-rich scenarios and (2) such an approach effectively helps to mitigate dimensional collapse in attribute-poor scenarios. However, we would also like to point out that even if LINKX successfully mitigates dimensional collapse for GNNs, it is not ideal, since

Table 9: Node classification accuracy comparison between GNN and APPNP on Cora dataset. Here we adopt the walk-based GCN as a representative GNN for comparison, and both models are optimized with the walk-based loss.

|  | 100%Att, Acc | 100%Att, EDR | 1%Att, Acc | 1%Att, EDR | 0.01%Att, Acc | 0.01%Att, EDR |
|---|---|---|---|---|---|---|
| **GCN** | 67.8% | 96.9% | 36.9% | 35.5% | 32.3% | 1.56% |
| **APPNP** | 75.5% | 56.3% | 42.4% | 13.7% | 10.6% | 0.78% |

Table 10: Node classification accuracy comparison between GNN and LINKX on Cora dataset. Here we adopt the walk-based GCN as a representative GNN for comparison, and both models are optimized with the walk-based loss.

|  | 100%Att, Acc | 100%Att, EDR | 1%Att, Acc | 1%Att, EDR | 0.01%Att, Acc | 0.01%Att, EDR |
|---|---|---|---|---|---|---|
| **GCN** | 67.8% | 96.9% | 36.9% | 35.5% | 32.3% | 1.56% |
| **LINKX** | 65.9% | 50.4% | 64.6% | 49.6% | 68.8% | 24.2% |

it (1) sacrifices the capability of GNNs in inductive learning and (2) improves the computational complexity from $\mathcal{O}(n * k)$ to $\mathcal{O}(n^2)$ to perform inference ($k$ is the number of node attributes and $n$ is the number of nodes). Therefore, the problem of dimensional collapse is non-trivial to handle, and more analysis can be a great follow-up study of our work.

## C.11  DISCUSSION: WORKS COMBINING SHALLOW METHODS AND GNNS

Here we present two representative works that aim to combine the advantage of shallow graph embedding methods and GNNs (Abu-El-Haija et al., 2018; Chien et al., 2020). Specifically, (Abu-El-Haija et al., 2018) successfully achieves optimization for the context hyper-parameters of shallow graph embedding methods. However, the proposed approach cannot take node attributes as the input and thus fails to effectively utilize the information encoded in node attributes; (Chien et al., 2020) explored to generalize the GNNs to adaptively learn high-quality node representations under both homophilic and heterophilic node label patterns. Nevertheless, it fails to avoid using the node attributes as the learning prior, and thus still follows a design that has been proved to bear dimensional collapse in this paper.

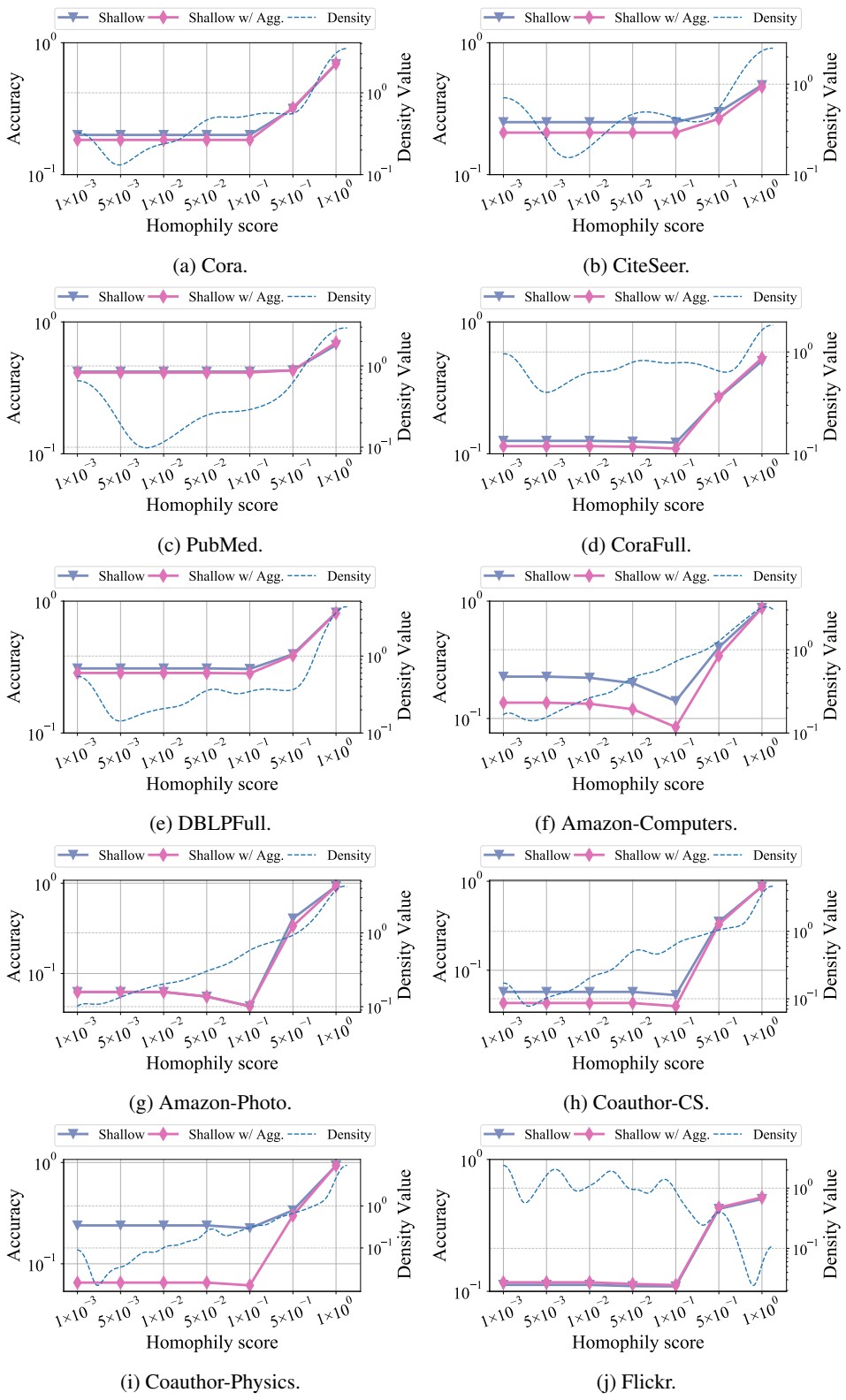

Figure 13: A comparison between shallow embedding methods and shallow embedding methods with an enforced neighborhood aggregation on 10 real-world graph datasets Here performance is measured with the cumulative node classification accuracy, and the density curve (marked with dashed line) represents the density of nodes with a certain homophily score in the test set.

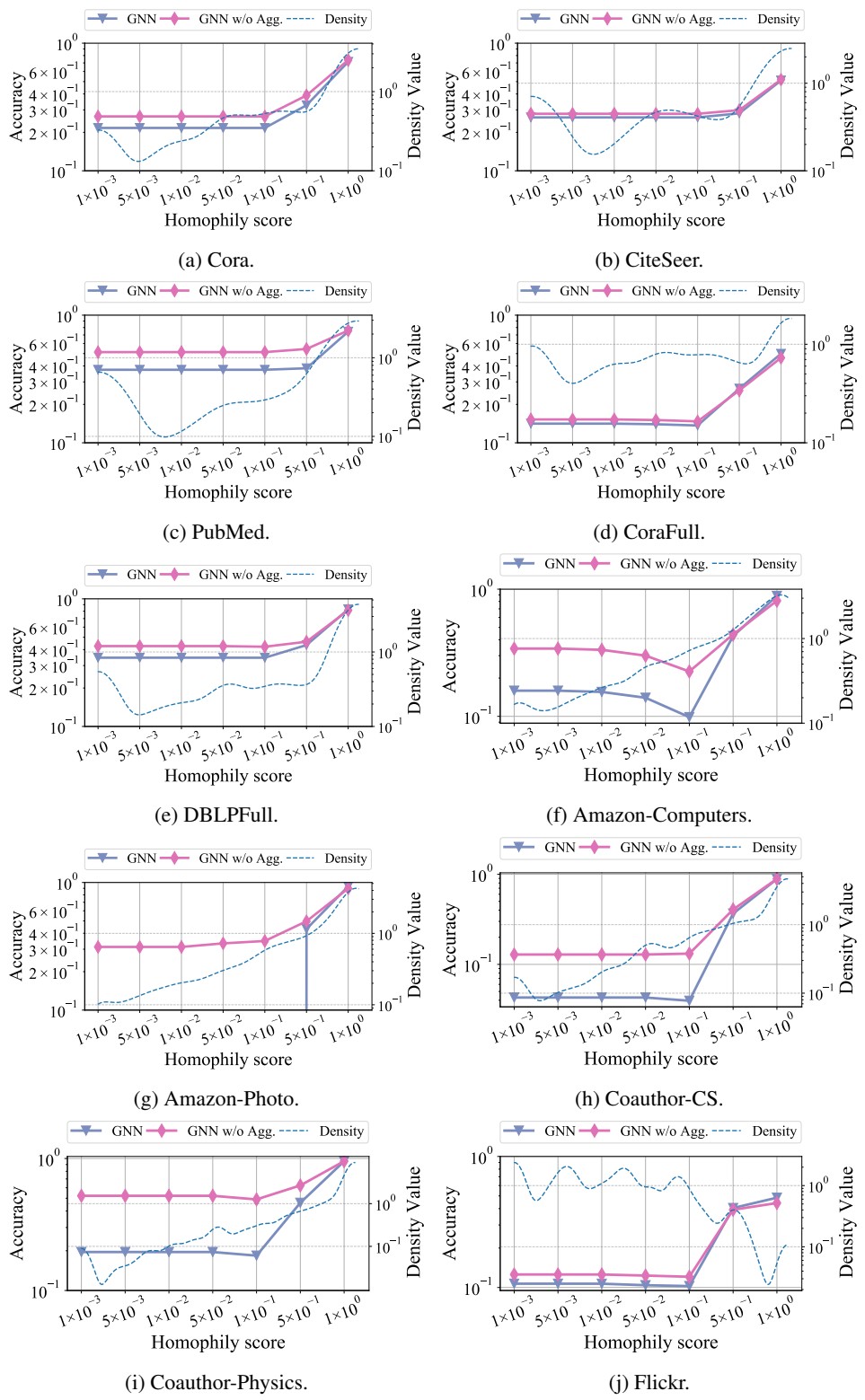

(a) Cora.  (b) CiteSeer.

(c) PubMed.  (d) CoraFull.

(e) DBLPFull.  (f) Amazon-Computers.

(g) Amazon-Photo.  (h) Coauthor-CS.

(i) Coauthor-Physics.  (j) Flickr.

Figure 14: A comparison between GNNs and GNNs without the neighborhood aggregation on 10 real-world graph datasets Here performance is measured with the cumulative node classification accuracy, and the density curve (marked with dashed line) represents the density of nodes with a certain homophily score in the test set.

