# OpenReview forum: "SEESAW: Do Graph Neural Networks Improve Node Representation Learning for All?"
_ICLR.cc/2024/Conference — Submitted to ICLR 2024_

### Official Review · Reviewer_ySnn · 2023-10-21

**Soundness:** 3 good
**Presentation:** 3 good
**Contribution:** 2 fair
**Rating:** 5
**Confidence:** 4

**Summary:**

This paper proposes a framework that unifies shallow graph embedding methods and Graph Neural Networks (GNNs) for node representation learning. The authors conduct a comparative analysis of the primary differences in design between the two approaches from the perspectives of node representation learning and neighborhood aggregation mechanism. Through comprehensive experiments on ten real-world graph datasets, the authors provide insights into the benefits and drawbacks of using GNNs and propose a guide for practitioners in choosing appropriate graph representation learning models under different scenarios. The paper aims to provide a broader perspective on graph learning and to recalibrate the academic perspective on the effectiveness of GNNs compared to conventional shallow embedding methods.

**Strengths:**

1. The author meticulously elaborated step by step on the relationship between prediction performance drop of GNNs and attribute dimension, as well as dimension collapse in hidden space,  which is sound.
2. The paper helps clarify the respective strengths and weaknesses of GNN and shallow embedding methods, making it a valuable reference for practitioners.

**Weaknesses:**

1. This article lacks novelty to some extent, despite providing detailed analysis and experiments. The conclusions are relatively trivial and are already a consensus in the community.
2. The paper may be improved if it discusses some works that combine the advantage of network embedding and GNN, like [1,2].

[1] Abu-El-Haija S, Perozzi B, Al-Rfou R, et al. Watch your step: Learning node embeddings via graph attention[J]. Advances in neural information processing systems, 2018, 31.
[2] Chien E, Peng J, Li P, et al. Adaptive Universal Generalized PageRank Graph Neural Network[C]//International Conference on Learning Representations. 2020.

**Questions:**

(1) The result of E2E-GCN on GCN, CiteSeer, Pubmed is lower than that reported in the paper. Can the authors explain the difference of experimal setting?
(2) Since the GNNs usually contain non-linear activation functions, is it reasonable to measure the Dimensional Collapse by evaluating rank of the embedding matrix?
(3) Is it possible to overcome the weakness of both GNNs and shallow embedding methods, and propose a new graph representation paradigm to combine their strengths?

---

> ### Author Response · Authors · 2023-11-22
> **Author Response to Reviewer ySnn (1/2)**
>
> We sincerely appreciate the time and efforts you've dedicated to reviewing and providing invaluable feedback to enhance the quality of this paper. We provide a point-to-point reply below for the mentioned concerns and questions.
>
> ---
>
>
> >  **Reviewer**: W1. This article lacks novelty to some extent, despite providing detailed analysis and experiments. The conclusions are relatively trivial and are already a consensus in the community.
>
> **Authors**: We agree with the reviewer that a part of the observations revealed in this paper has also been reported by existing literatures, e.g., GNNs could bear sub-optimal performances on highly heterophilic nodes. However, we would like to note that the main focus of this paper is to take an initial step to perform a comprehensive comparison between GNNs and shallow methods. Especially, to the best of our knowledge, no existing work has comprehensively explored the strengths and weaknesses of GNNs compared with shallow methods with a unified view. Therefore, a comprehensive analysis presented in our paper is still needed to have an in-depth understanding of GNNs. For example, it is valuable to point out that dimensional collapse solely happens in GNNs instead of traditional shallow methods, and such a conclusion has not been revealed by any other existing works.
>
> ---
>
>
> >  **Reviewer**: W2. The paper may be improved if it discusses some works that combine the advantage of network embedding and GNN, like [1,2].
>
> **Authors**: We would like to thank the reviewer for pointing this out. Specifically, we note that [1] successfully achieves optimization for the context hyper-parameters of shallow graph embedding methods. However, the proposed approach cannot take node attributes as the input and thus fails to effectively utilize the information encoded in node attributes; [2] explored to generalize the GNNs to adaptively learn high-quality node representations under both homophilic and heterophilic node label patterns. Nevertheless, it fails to avoid using the node attributes as the learning prior, and thus still follows a design that has been proved to bear dimensional collapse in our paper.
>
> We thank the reviewer again for bringing this up so we can make our discussion more comprehensive. We have included the discussion in Appendix C.11.
>
> [1] Abu-El-Haija S, Perozzi B, Al-Rfou R, et al. Watch your step: Learning node embeddings via graph attention[J]. Advances in neural information processing systems, 2018, 31.
>
> [2] Chien E, Peng J, Li P, et al. Adaptive Universal Generalized PageRank Graph Neural Network[C]//International Conference on Learning Representations. 2020.
>
> ---
>
> >  **Reviewer**: (1) The result of E2E-GCN on GCN, CiteSeer, Pubmed is lower than that reported in the paper. Can the authors explain the difference of experimal setting?
>
> **Authors**: We thank the reviewer for pointing this out. We utilized standard implementations of the GCN layer from standard packages, and the performance differences do not influence our observations and conclusions.  We agree with the reviewer that such inconsistency over these most commonly used datasets could lead to confusion, and we have changed the corresponding performances to the results obtained based on the implementation from the original papers to avoid any further confusion.

---

> ### Author Response · Authors · 2023-11-22
> **Author Response to Reviewer ySnn (2/2)**
>
> ---
>
> >  **Reviewer**: (2) Since the GNNs usually contain non-linear activation functions, is it reasonable to measure the Dimensional Collapse by evaluating rank of the embedding matrix?
>
> **Authors**: We would like to argue that (1) it is **reasonable to measure** the Dimensional Collapse with the rank of the representation matrix and (2) whether a model contains non-linear activation functions or not **does not influence the choice of metrics** to measure the Dimensional Collapse. We elaborate on the details below.
>
> First, Dimensional Collapse refers to the phenomenon where the representations of data points (i.e., the input nodes in our paper) collapse into a lower-dimensional subspace instead of spanning the entire available hidden space. Accordingly, the rank of the representation matrix directly reveals the dimensionality of the subspace these representations span, which is also consistent with a series of recent works (e.g., [1, 2]). Therefore, we argue that it is **reasonable to measure** the Dimensional Collapse with the rank of the representation matrix.
>
> Second, we note that the major connection between non-linear activation functions and Dimensional Collapse is that when non-linear activation functions are adopted in a model, they may improve the rank of the output representation matrix (compared with the input feature matrix) and thus may mitigate the level of Dimensional Collapse (we have empirically found that such mitigation is marginal in GNNs). Such a connection does not influence the semantic meaning of the rank of the representation matrix, and thus **does not influence the appropriate choice of metrics** to measure the Dimensional Collapse.
>
> [1] Roth, A., & Liebig, T. (2023). Rank Collapse Causes Over-Smoothing and Over-Correlation in Graph Neural Networks. arXiv preprint arXiv:2308.16800.
>
> [2] Sun, J., Chen, R., Li, J., Wu, C., Ding, Y., & Yan, J. (2022). On Understanding and Mitigating the Dimensional Collapse of Graph Contrastive Learning: a Non-Maximum Removal Approach. *arXiv preprint arXiv:2203.12821*.
>
> ---
>
> >  **Reviewer**: (3) Is it possible to overcome the weakness of both GNNs and shallow embedding methods, and propose a new graph representation paradigm to combine their strengths?
>
> **Authors**: We agree with the reviewer that there could be ways to combine their strengths, and we believe this would be an interesting future direction to work on. However, despite the significance of this problem, we would also like to note that **handling such a problem is difficult**. In fact, most existing works fail to avoid the weaknesses characterized by our paper, and thus they are not able to properly combine the strengths of the two methods. We believe it would be exciting for the graph machine learning community to explore such a new future direction.
>
> In fact, it is great that this paper can lead to questions like this from the readers since we believe that this is part of what we want with this work: to have researchers and practitioners not only looking at GNNs but also traditional shallow embedding methods for further advancements.

---

> > ### Comment · Reviewer_ySnn · 2023-11-22
> >
> > Thanks for your detailed response.
> >
> > I think the rethinking and comparison of GNNs with shallow models is valuable, and the analysis of dimension collapse for GNNs is insightful.  However, I believe this article would be improved if it proposed new methods to address the challenges of GNN/shallow models, or make some cases of applications in some new algorithm that would benefit from both paradigms. This is indeed a difficult and valuable direction. For now, I think its current status does not meet the standards of ICLR. Therefore, I would keep my score for now.

---

> > > ### Author Response · Authors · 2023-11-22
> > > **Author Reply to Reviewer ySnn**
> > >
> > > Thank you for your prompt reply, and we truly appreciate your recognition.
> > >
> > > We agree that being able to propose a state-of-the-art solution to solve this problem would be great. Nonetheless, this would require huge efforts that can easily go beyond the scope of a single paper, especially considering that our paper already contains 20+ pages. We'd like to note that one of the main goals of this paper is to carefully pose this problem with a systematic analysis and draw more attention from the research community to it, as the main research interests have been shifting away from the traditional shallow methods in the past few years. We argue that these efforts make our work particularly interesting to the graph learning community, and our study is also critical for any future follow-up works that could propose a SOTA solution to it.

---

### Official Review · Reviewer_4Gbd · 2023-10-22

**Soundness:** 2 fair
**Presentation:** 3 good
**Contribution:** 2 fair
**Rating:** 5
**Confidence:** 3

**Summary:**

This paper scrutinizes drawbacks of GNNs compared to shallow embedding approaches. Specifically, the authors observe that GNNs suffer dimensional collapse and exhibit poor performance when input features are limited. Furthermore, in heterophilic graphs, GNNs with aggregation also show inferior performance compared to GNNs without aggregation and shallow embedding methods. Considering these observations, they suggest to use GNNs when input attributes are rich, graphs are homophilic, transductive or large.

**Strengths:**

* The paper is well-written.
* The authors validate their hypothesis on various datasets.

**Weaknesses:**

* It is already shown that MLPs (which is same with GNNs without aggregation) outperforms commonly used GNNs such as GCN, GAT, and SAGE. Furthermore, several methods are proposed to perform well on both homophilous and heterophilous graphs [1, 2, 3].
* It seems that the circumstances where input features are limited is not common in the real-world senarios and we can augment input features with large language models even in attribute-poor settings [4].
* It is not surprising that GNNs suffer dimensional collapses when input features are poor since it is well-known for general neural netoworks.

[1] Zhu, Jiong, et al. "Beyond homophily in graph neural networks: Current limitations and effective designs." Advances in neural information processing systems 33 (2020): 7793-7804.

[2] Lim, Derek, et al. "Large scale learning on non-homophilous graphs: New benchmarks and strong simple methods." Advances in Neural Information Processing Systems 34 (2021): 20887-20902.

[3] Yang, Liang, et al. "Diverse message passing for attribute with heterophily." Advances in Neural Information Processing Systems 34 (2021): 4751-4763.

[4] Xiaoxin He, et al. "Harnessing Explanations: LLM-to-LM Interpreter for Enhanced Text-Attributed Graph Representation Learning." arXiv (2023).

**Questions:**

* There are several models combining walking-based approaches and GNNs such as APPNP [1]. I think that this kind of mechanism might alleviate the problem of GNNs due to the adoption of pagerank. Does APPNP also suffer similar problems as other GNNs?
* Several approaches such as LINKX [2] encode node topology and node attribute separately and combine two representations later. Since these approaches can learn how much to reflect node attributes on node representations, I think that these methods might not suffer dimensional collapse. Does LINKX also suffer similar problems?

[1] Gasteiger, Johannes, Aleksandar Bojchevski, and Stephan Günnemann. "Predict then propagate: Graph neural networks meet personalized pagerank." arXiv preprint arXiv:1810.05997 (2018).

[2] Lim, Derek, et al. "Large scale learning on non-homophilous graphs: New benchmarks and strong simple methods." Advances in Neural Information Processing Systems 34 (2021): 20887-20902.

---

> ### Author Response · Authors · 2023-11-22
> **Author Response to Reviewer 4Gbd (1/3)**
>
> We sincerely appreciate the time and efforts you've dedicated to reviewing and providing invaluable feedback to enhance the quality of this paper. We provide a point-to-point reply below for the mentioned concerns and questions.
>
> ---
>
>
> >  **Reviewer**: W1. It is already shown that MLPs (which is same with GNNs without aggregation) outperforms commonly used GNNs such as GCN, GAT, and SAGE. Furthermore, several methods are proposed to perform well on both homophilous and heterophilous graphs [1, 2, 3].
>
> **Authors**: We thank the reviewer for pointing this out. We have been aware that MLPs may outperform GNNs in certain cases, and our work is not tailored for the homophily level of the input graph data. We elaborate on the details below.
>
> We note that the main focus of this paper is to perform a comprehensive comparison between GNNs and shallow embedding methods. The observation and discussion will remain on both homophilous and heterophilous graphs. Existing works (e.g., [1, 2, 3]) do not explicitly explore the strengths and weaknesses of GNNs compared with shallow methods. We argue that the novelty of our paper mainly lies in the comprehensive comparison of (1) the levels of dimensional collapse and (2) performances over different levels of homophily between the two types of models.
>
> [1] Zhu, Jiong, et al. "Beyond homophily in graph neural networks: Current limitations and effective designs." Advances in neural information processing systems 33 (2020): 7793-7804.
>
> [2] Lim, Derek, et al. "Large scale learning on non-homophilous graphs: New benchmarks and strong simple methods." Advances in Neural Information Processing Systems 34 (2021): 20887-20902.
>
> [3] Yang, Liang, et al. "Diverse message passing for attribute with heterophily." Advances in Neural Information Processing Systems 34 (2021): 4751-4763.
>
> ---
>
>
> >  **Reviewer**: W2. It seems that the circumstances where input features are limited is not common in the real-world senarios and we can augment input features with large language models even in attribute-poor settings [4].
>
> **Authors**: We thank the reviewer for pointing this out. We would like to note that, in fact, the circumstances where input features are limited are common. For example, online social networks have become prevalent, while it is common that anonymized social networks usually lack the identity information of nodes [1]. As another example, the attribute of a node in social networks may also be an artifact and captures no semantic meanings [1]. Therefore, the phenomenons revealed by our paper widely exist, and such a weakness would also be shown across a series of real-world applications.
>
> We also agree with the reviewer that input features could be augmented with large language models in attribute-poor settings, which is an exciting future direction to explore. However, augmenting features with LLMs might not scale to real-world scenarios. For example, a social network could have billions of nodes [2], and generating node features with LLMs may bear high computational costs.
>
> [1] Sun, Z., Zhang, W., Mou, L., Zhu, Q., Xiong, Y., & Zhang, L. (2022, June). Generalized equivariance and preferential labeling for gnn node classification. In Proceedings of the AAAI Conference on Artificial Intelligence (Vol. 36, No. 8, pp. 8395-8403).
>
> [2] Przepiorka, A., & Blachnio, A. (2016). Time perspective in Internet and Facebook addiction. Computers in Human Behavior, 60, 13-18.
>
> ---
>
>
> >  **Reviewer**: W3. It is not surprising that GNNs suffer dimensional collapses when input features are poor since it is well-known for general neural netoworks.
>
> **Authors**: We agree with the reviewer that dimensional collapse has been found to happen in general neural networks by a series of existing works. However, we would like to point out that such a phenomenon does not necessarily mean that GNNs will also suffer a similar problem. Especially, to the best of our knowledge, no existing work has comprehensively explored the strengths and weaknesses of GNNs compared with shallow methods with a unified view. Therefore, a comprehensive analysis is still needed to have an in-depth analysis of GNNs. For example, it is valuable to point out that dimensional collapse solely happens in GNNs instead of traditional shallow methods, and such a conclusion has not been revealed by any other existing works.

---

> ### Author Response · Authors · 2023-11-22
> **Author Response to Reviewer 4Gbd (2/3)**
>
> ---
>
> >  **Reviewer**: There are several models combining walking-based approaches and GNNs such as APPNP [1]. I think that this kind of mechanism might alleviate the problem of GNNs due to the adoption of pagerank. Does APPNP also suffer similar problems as other GNNs?
>
> **Authors**: We thank the reviewer for pointing this out. First, we would like to point out that in the mentioned model APPNP, only the way of the information distribution in APPNP is similar to the random walk. However, as is revealed in our paper, **the problem of dimensional collapse is caused by utilizing the node attributes as the prior of learning**. Since APPNP still utilizes the node attributes as the prior of learning, **APPNP does not bear any significant difference in terms of the cause for dimensional collapse** compared with the GNNs adopted in this paper. Therefore, we conclude that APPNP naturally suffers similar problems as other GNNs.
>
> In addition, we also perform empirical experiments based on APPNP. We present the unsupervised learning performances on the Cora dataset below as an example. Here utility is measured by node classification accuracy, while the effective dimension ratio (EDR) is measured by the ratio of the value of rank (of representation matrix) to the representation dimensionality. We observe that similar to GCN, APPNP also bears severe dimensional collapse (exhibited by the significant reduction in the value of EDR).
>
> |       | 100%Att, Acc | 100%Att, EDR | 1%Att, Acc | 1%Att, EDR | 0.01%Att, Acc | 0.01%Att, EDR |
> | ----- | ------------ | ------------ | ---------- | ---------- | ------------- | ------------- |
> | GCN   | 67.8%        | 96.9%        | 36.9%      | 35.5%      | 32.3%         | 1.56%         |
> | APPNP | 75.5%        | 56.3%        | 42.4%      | 13.7%      | 10.6%         | 0.78%         |
>
> We note that the discussion above reveals that the problem pointed out by our work is non-trivial to handle. Correspondingly, this paper is particularly interesting to researchers working in this area and follow-up studies of our work remain desired.
>
> We thank the reviewer again for bringing this up so we can make our evaluation more comprehensive. We have included the experimental results and the corresponding discussion in Appendix C.10.

---

> ### Author Response · Authors · 2023-11-22
> **Author Response to Reviewer 4Gbd (3/3)**
>
> ---
>
> >  **Reviewer**: Several approaches such as LINKX [2] encode node topology and node attribute separately and combine two representations later. Since these approaches can learn how much to reflect node attributes on node representations, I think that these methods might not suffer dimensional collapse. Does LINKX also suffer similar problems?
>
> **Authors**: We agree with the reviewer that dimensional collapse could be mitigated by encoding node topology and node attribute separately and combining two representations later. However, it is also worth noting that **the adjacency matrix is naturally low-rank as well** [1, 2], and thus it could be difficult to avoid dimensional collapse with the operations above. Specifically, we perform empirical experiments based on LINKX. We present the unsupervised learning performances on the Cora dataset below as an example. Here utility is measured by node classification accuracy, while the effective dimension ratio (EDR) is measured by the ratio of the value of rank (of representation matrix) to the representation dimensionality. We observe that compared with GCN, LINKX exhibits smaller values of EDR in attribute-rich scenarios (e.g., 100% available node attributes), while it also mitigates dimensional collapse in attribute-poor scenarios (e.g., 0.01% available node attributes). This demonstrates that (1) such an approach may jeopardize the effective dimension ratio in attribute-rich scenarios and (2) such an approach effectively helps to mitigate dimensional collapse in attribute-poor scenarios.
>
> |       | 100%Att, Acc | 100%Att, EDR | 1%Att, Acc | 1%Att, EDR | 0.01%Att, Acc | 0.01%Att, EDR |
> | ----- | ------------ | ------------ | ---------- | ---------- | ------------- | ------------- |
> | GCN   | 67.8%        | 96.9%        | 36.9%      | 35.5%      | 32.3%         | 1.56%         |
> | LINKX | 65.9%        | 50.4%        | 64.6%      | 49.6%      | 68.8%         | 24.2%         |
>
> However, we would also like to point out that even if such a method successfully mitigates dimensional collapse for GNNs, it **is not ideal**, since it (1) sacrifices the capability of GNNs in inductive learning and (2) improves the computational complexity from $\mathcal{O}(n*k)$  to $\mathcal{O}(n^2)$ to perform inference ($k$ is the number of node attributes and $n$ is the number of nodes). Therefore, the problem of dimensional collapse is non-trivial to handle, and more analysis can be a great follow-up study of our work.
>
> We thank the reviewer again for bringing this up so we can make our evaluation more comprehensive. We have included the experimental results and the corresponding discussion in Appendix C.10.
>
> [1] Entezari, N., Al-Sayouri, S. A., Darvishzadeh, A., & Papalexakis, E. E. (2020, January). All you need is low (rank) defending against adversarial attacks on graphs. In *Proceedings of the 13th International Conference on Web Search and Data Mining* (pp. 169-177).
>
> [2] Zhuang, L., Gao, H., Lin, Z., Ma, Y., Zhang, X., & Yu, N. (2012, June). Non-negative low rank and sparse graph for semi-supervised learning. In *2012 ieee conference on computer vision and pattern recognition* (pp. 2328-2335). IEEE.

---

> ### Comment · Reviewer_4Gbd · 2023-11-22
> **Further concern**
>
> I appreciate the detailed responses and several of my concerns are addressed. However, upon reviewing the results, I have an additional concern. I think that LINKX serves as a compelling counter-example to one of the main assertions made in the paper. Specifically, the authors show that many Graph Neural Networks (GNNs) suffer a dimensional collapse in attribute-poor settings, and this collapse is linked to performance in observations. Nevertheless, LINKX effectively mitigates dimensional collapse and demonstrates consistent performance across various ratios of accessible attributes, even as EDR decreases. Consequently, the necessity of utilizing the representation concatenation of GNNs and shallow models (a natural extension from the findings) seems weak, since LINKX can successfully mitigate this issue, leading to a reduction in the broad effect of the finding.  It would be more convincing to provide evidence exhibiting the superiority of the proposed approach over LINKX.

---

> > ### Author Response · Authors · 2023-11-22
> > **Reply to Reviewer 4Gbd**
> >
> > We appreciate your prompt feedback and recognition.
> >
> > We agree with the reviewer that LINKX effectively mitigates dimensional collapse and demonstrates consistent performance. However, we would like to clarify the misunderstanding here: (1) utilizing the concatenation of the representations from the two methods **only serves as an example** of combining the two methods; (2) we **do not argue** this concatenation-based approach should be adopted instead of other potential possibilities, and we **do not argue for its superiority** either; (3) the main contribution of this paper lies in the comprehensive comparison between the two types of models. In fact, we are excited to see that insights can be brought out by our initial explorations, which is also one of the main purposes of our paper.
> >
> > We hope the clarification above helps to address your concern. We thank the reviewer again for the constructive feedback, and we are also eager to engage in further discussion.

---

> ### Author Response · Authors · 2023-11-22
> **A kind reminder**
>
> Dear Reviewer 4Gbd,
>
> We would like to express our sincere gratitude to you for reviewing our paper and providing valuable feedback. We believe that we have responded to and addressed all your concerns with our revisions — in light of this, we hope you consider raising your score.
>
> Notably, given that we are approaching the deadline for the rebuttal phase, we hope we can have the discussion soon. Thanks！
>
> Best,
> All authors

---

> > ### Comment · Reviewer_4Gbd · 2023-11-23
> > **Change my score**
> >
> > Thank you for your detailed response. Many concerns have been addressed, leading me to raise the score to a marginal acceptance. Since some of the recent GNNs mitigate the issue and I feel that the broad effect of the finding seems limited, I decided on this score.

---

> ### Author Response · Authors · 2023-11-23
> **Reply to Reviewer 4Gbd**
>
> Thank you for your prompt reply, and we truly appreciate your recognition.
>
> We agree that a few of the recent GNNs mitigate this issue, but they also bear their major weaknesses (e.g., sacrificing inductive learning capability for LINKX) and thus are not ideal strategies. In light of this, we would like to note that one of the main contributions of this paper is to carefully pose this problem with a systematic analysis and draw more attention from the research community to it, as the main research interests have been shifting away from the traditional shallow methods in the past few years. We argue that these efforts make our work particularly interesting to the graph learning community to exert a broader impact, and our study is also critical for any future follow-up works that could propose better solutions to handle the critical weaknesses of GNNs identified in our paper. We hope this addresses your concern about the broad impact of our paper.

---

### Official Review · Reviewer_4Dj2 · 2023-10-31

**Soundness:** 3 good
**Presentation:** 3 good
**Contribution:** 2 fair
**Rating:** 5
**Confidence:** 2

**Summary:**

This paper compares classic graph embedding methods (e.g., DeepWalk) and GNNs. For this, the authors first unify the setups: namely, GNN optimizes the same random walk objective. The paper argues that there are two main differences between GNNs and classic approaches: different prior representations and different updating operations (whether there is aggregation over the neighbors).

In the experiments the following observations are made:
- On one out of ten datasets DeepWalk outperforms GNNs;
- The performance of GNNs drops when features are removed;
- For GNNs, the dimensionality of learned representations decreases when features are removed;
- The performance of GNNs and shallow models is better for high-homophily nodes. For low-homophily nodes removing neighborhood aggregation improves the performance.

Based on the conducted experiments, suggestions for when it is better to use which model are given.

**Strengths:**

1. The paper addresses an important topic.

2. Extensive experiments are conducted to analyze and compare the performance of GNNs and classic methods.

3. The paper is in general clearly written and easy to follow.

**Weaknesses:**

1. The novelty of the work seems limited - most of the observations are straightforward or appeared in previous research.

2. While the paper contains a guide for practitioners about which model to choose, it is not specific to be directly applied to a given application. For instance, in Section 5, it is written "we recommend adopting GNNs and shallow embedding methods on attribute-rich and attribute-poor networks, respeectively." However, it is not clear how to decide whether the attributes are rich. For instance, in both Flickr and PubMed, there are 500 features, but the results on them are completely different. So, it is not the number of features that can be used for this decision.

**Questions:**

Q1. How to decide whether the features are sufficiently rich?

Q2. It is written that "It is difficult for shallow embedding methods to properly exploit information encoded in node attributes and make use of the homophily nature of most graphs" - why the latter is true? Classic methods have similar embeddings for nodes located close to each other in the graph. Under the homophily assumption, such nodes often have the same label.

Q3. The fact that GNNs strongly rely on node features and removing them leads to decreased performance is very natural. Can this problem be solved by augmenting node features with structural graph-based features? Or maybe even with random features? Both options can also increase the representation effective dimension.

Q4. Can small representation effective dimension be explained just by the dimension of the initial feature set? This would also explain Figure 5(b) since increased rank bound cannot solve this issue.

Q5. The concatenation experiment is conducted only on one dataset (DBLPFull). Are the results on other datasets consistent with this?

Q6. Do I understand correctly that GNN w/o Agg (Figure 6) does not use any graph structure?

There are some typos throughout the text:
- Page 2: "the most graph popular representation"
- Page 2: footnote is placed before the punctuation mark
- Page 3: (line 2, line 3): "output" -> "outputs"
- Page 3: "There have been various of GNNs"
- Section 4.2: "We hypothesis that"
- Section 4.2: "methods preserves"
- Section 4.2, page 6: in the definition of matrices Z, C, F, the matrix Z repeats twice (same typo in appendix, page 17)
- Page 9: "respeectively"
- Page 20: "the available node attributes becomes"

---

> ### Author Response · Authors · 2023-11-22
> **Author Response to Reviewer 4Dj2 (1/3)**
>
> We sincerely appreciate the time and efforts you've dedicated to reviewing and providing invaluable feedback to enhance the quality of this paper. We provide a point-to-point reply below for the mentioned concerns and questions.
>
> ---
>
>
> >  **Reviewer**: W1. The novelty of the work seems limited - most of the observations are straightforward or appeared in previous research.
>
> **Authors**: We agree with the reviewer that a part of our observations has been revealed by existing works.  However, we note that the main focus of this paper is to perform a comprehensive comparison between GNNs and shallow embedding methods. Especially, to the best of our knowledge, no existing work has comprehensively explored the strengths and weaknesses of GNNs compared with shallow methods with a unified view. Therefore, a comprehensive analysis is still needed to have an in-depth analysis of GNNs. For example, it is valuable to point out that dimensional collapse solely happens in GNNs instead of traditional shallow methods, and such a conclusion has not been revealed by any other existing works.
>
> ---
>
>
> >  **Reviewer**: W2. While the paper contains a guide for practitioners about which model to choose, it is not specific to be directly applied to a given application. For instance, in Section 5, it is written "we recommend adopting GNNs and shallow embedding methods on attribute-rich and attribute-poor networks, respeectively." However, it is not clear how to decide whether the attributes are rich. For instance, in both Flickr and PubMed, there are 500 features, but the results on them are completely different. So, it is not the number of features that can be used for this decision.
>
> **Authors**: We thank the reviewer for pointing this out. We would like to note that rich attributes do not necessarily mean a large number of available features. For example, no matter what the dimensionality of the input node features is, if most of the available features are linearly correlated with each other, i.e., the rank of the input feature matrix is small, then dimensional collapse is still likely to happen. This is because as shown in Figure 5.b, the non-linear operations in GNNs can hardly improve the rank of representations (compared with node features) regardless of the dimensionality. Correspondingly, we note that it is the rank of the input features that truly determines whether it is an attribute-poor scenario or not. However, since the rank of the input feature matrix cannot exceed the number of attribute dimensions, the number of attribute dimensions also plays a critical role in determining whether it is an attribute-poor scenario. In addition, we note that both Flickr and PubMed are not attribute-poor datasets, since the GNN is able to fully utilize most hidden dimensionalities when 100% attributes are available. We have improved the corresponding explanation in Section 4.3 and Section 5 accordingly to avoid further confusion.
>
> Furthermore, we would like to point out that it is not appropriate to compare performances across different datasets, since how much the node attributes help with the GNN prediction could vary across different datasets. Instead, we present Figure 2 to compare the performances within each dataset under different levels of available input attribute dimensions, which makes the analysis of dimensional collapse more rigorous.
>
> ---
>
> >  **Reviewer**: Q1. How to decide whether the features are sufficiently rich?
>
> **Authors**: We thank the reviewer for pointing this out. We would like to note that whether the features are sufficiently rich or not **is usually determined by the specific downstream task**, which is also the reason why there is **no unified metric** to measure whether the features are "rich" enough. For example, in a node regression task (i.e., predict a specific value regarding each node), we may not need the embeddings to span the whole hidden space, since more discriminative node embeddings may not be as helpful in improving the performance as in node classification tasks [1]. Accordingly, the requirement in the "richness" of node features could be less strict compared with that for node classification tasks.
>
> To avoid further confusion and misunderstanding, we added a discussion on whether the features of specific graph data are sufficiently rich or not in our Appendix. However, this goes beyond the scope of this paper of performing a comprehensive comparison between the two types of models.
>
> [1] Yu, Y., Chan, K. H. R., You, C., Song, C., & Ma, Y. (2020). Learning diverse and discriminative representations via the principle of maximal coding rate reduction. Advances in Neural Information Processing Systems, 33, 9422-9434.

---

> ### Author Response · Authors · 2023-11-22
> **Author Response to Reviewer 4Dj2 (2/3)**
>
> ---
>
> >  **Reviewer**: Q2. It is written that "It is difficult for shallow embedding methods to properly exploit information encoded in node attributes and make use of the homophily nature of most graphs" - why the latter is true? Classic methods have similar embeddings for nodes located close to each other in the graph. Under the homophily assumption, such nodes often have the same label.
>
> **Authors**: We would like to thank the reviewer for pointing this out. We agree that the latter is not appropriate, and the main claim here is that shallow embedding methods cannot effectively exploit node attribute information. We have revised the expression accordingly to avoid confusion.
>
> ---
>
> >  **Reviewer**: Q3. The fact that GNNs strongly rely on node features and removing them leads to decreased performance is very natural. Can this problem be solved by augmenting node features with structural graph-based features? Or maybe even with random features? Both options can also increase the representation effective dimension.
>
> **Authors**: We agree with the reviewer that feature augmentation could be a potential direction to explore, such that the disadvantage of GNNs revealed in this paper can be mitigated. However, the problem of dimensional collapse cannot be properly addressed simply by augmenting node attributes by either adding structural graph-based features or adding random features. We performed experiments by (1) concatenating a random matrix with the same dimensionality as the original node attributes onto the node attribute matrix and (2) concatenating a matrix encoded with structural information following the state-of-the-art *degree+* strategy [1] onto the node attribute matrix. We present the unsupervised learning performances on the Cora dataset below as an example. Here utility is measured by node classification accuracy, while the effective dimension ratio (EDR) is measured by the ratio of the value of rank (of representation matrix) to the representation dimensionality.
>
> |                      | 100%Att, Acc | 100%Att, EDR | 1%Att, Acc | 1%Att, EDR | 0.01%Att, Acc | 0.01%Att, EDR |
> | -------------------- | ------------ | ------------ | ---------- | ---------- | ------------- | ------------- |
> | Walk-GCN             | 67.8%        | 96.9%        | 36.9%      | 35.5%      | 32.3%         | 1.56%         |
> | Walk-GCN, Random     | 68.0%        | 97.3%        | 29.9%      | 67.2%      | 10.6%         | 5.08%         |
> | Walk-GCN, Structural | 69.0%        | 97.3%        | 37.3%      | 35.2%      | 32.3%         | 2.34%         |
>
> We observe that: (1) concatenating a matrix with structural information slightly improves the node classification accuracy, while such a strategy does not stop the significant drop in the rank of node representations; (2) concatenating random node attributes successfully improves the rank of the node representations, however, the classification accuracy is reduced. Therefore, both strategy does not really solve the problem of dimensional collapse, and we believe handling such a problem is non-trivial. Correspondingly, this paper is particularly interesting to researchers working in this area and such a problem is also worth to be explored in future works.
>
> We thank the reviewer again for bringing this up so we can make our evaluation more comprehensive. We have included the experimental results and the corresponding discussion in Appendix C.9.
>
> [1] Cui, H., Lu, Z., Li, P., & Yang, C. (2022, October). On positional and structural node features for graph neural networks on non-attributed graphs. In Proceedings of the 31st ACM International Conference on Information & Knowledge Management (pp. 3898-3902).
>
> ---
>
> >  **Reviewer**: Q4. Can small representation effective dimension be explained just by the dimension of the initial feature set? This would also explain Figure 5(b) since increased rank bound cannot solve this issue.
>
> **Authors**: We thank the reviewer for pointing this out and we agree with the reviewer. Specifically, if the number of the dimensions of the initial feature set is already small, this means the initial feature set bears the problem of low rank, since its rank cannot exceed its feature dimensionality. Accordingly, the learned node representations are then more likely to bear the dimensional collapse, since the GNNs cannot effectively improve the rank again even if the rank bound increases (i.e., the number of hidden dimensionality increases). Therefore, the number of attribute dimensions also plays a critical role in determining whether it is an attribute-poor scenario, and this is the reason why we adopt the ratio of the available number of node attributes to establish different levels of attribute richness. We have also added the associated explanation in Section 4.3 and Section 5 to make the discussion of our paper more comprehensive.

---

> ### Author Response · Authors · 2023-11-22
> **Author Response to Reviewer 4Dj2 (3/3)**
>
> ---
>
> >  **Reviewer**: Q5. The concatenation experiment is conducted only on one dataset (DBLPFull). Are the results on other datasets consistent with this?
>
> **Authors**: We thank the reviewer for pointing this out. We would like to note that for the results reported in this paper, we also have similar observations on other adopted datasets. In fact, we have presented the rank of representations learned by GNNs in Figure 9 (Appendix) and the rank of representations learned by shallow methods in Figure 11 (Appendix). The rank of the concatenation of two representation matrices will be at least the rank of the maximum rank of the two matrices. Noticing that in all datasets, the representations learned by shallow methods are full-rank. Therefore, the tendency curve of the concatenated representation will either reach a minimum of effective dimension ratio of 0.5 (similar to the results presented in Figure 8) or a value of effective dimension ratio better than 0.5 (i.e., greater than 0.5). We have also modified Section 5 for further clarification.
>
> ---
>
> >  **Reviewer**: Q6. Do I understand correctly that GNN w/o Agg (Figure 6) does not use any graph structure?
>
> **Authors**: Yes. For GNN w/o Agg, we remove all message-passing operations.
>
> ---
>
> >  **Reviewer**: There are some typos throughout the text.
>
> **Authors**: We thank the reviewer for pointing this out. We have revised them throughout this paper. Thanks again for your efforts to make this paper more enjoyable to read!

---

> > ### Comment · Reviewer_4Dj2 · 2023-11-23
> >
> > Thank you very much for your detailed feedback! It clarified some of my concerns. Also, I think that the new experiments strengthen the paper. However, at this point, I will keep my score unchanged. Comparing two classes of models in a unified setup is a strong point of the paper, but the overall contribution seems to be not sufficient for publication since many of the observations are either known or straightforward. From my point of view, the main takeaway from the paper is a guide for practitioners in choosing an appropriate graph model. However, the recommendations in Section 5 are very general, and it seems that reliable advice would be "It is worth trying both approaches because both of them have their advantages". I fully agree with the authors that having a definite answer to this question is probably impossible, but for this paper, this part seems to be the main motivation for conducting the analysis.

---

> ### Author Response · Authors · 2023-11-22
> **A kind reminder**
>
> Dear Reviewer 4Dj2,
>
> We would like to express our sincere gratitude to you for reviewing our paper and providing valuable feedback. We believe that we have responded to and addressed all your concerns with our revisions — in light of this, we hope you consider raising your score.
>
> Notably, given that we are approaching the deadline for the rebuttal phase, we hope we can have the discussion soon. Thanks！
>
> Best,
> All authors

---

### Official Review · Reviewer_VyuR · 2023-11-01

**Soundness:** 2 fair
**Presentation:** 4 excellent
**Contribution:** 2 fair
**Rating:** 3
**Confidence:** 4

**Summary:**

The work aims to compare the performance of GNNs and shallow embedding methods and delves into scenarios where GNNs may not always outperform shallow embedding methods. The authors present a systematic framework, SEESAW, to compare these two approaches. They identify key differences in learning priors and neighborhood aggregation and analyze when GNNs exhibit drawbacks. The study finds that GNNs may struggle in (1) attribute-poor scenarios, leading to dimensional collapse, and can adversely affect the performance of specific node subgroups in certain cases; (2) highly heterophilic networks, as the neighborhood aggregation may jeopardize the performance of heterophilic nodes. Thus, this paper suggests that practitioners should consider shallow embedding methods in attribute-poor scenarios and networks with heterophilic nodes.

**Strengths:**

1. This paper is well-structured and easy to follow.
2. The topic of comparing shallow embedding methods and GNN methods for graph representation learning is meaningful.
3. The experimental setting is clear, and code is provided.

**Weaknesses:**

1. Novelty is not good. The findings that GNNs face some challenges when we do not have enough attribute information (e.g.[1][2]) and when we have heterophilic data (e.g.[3]) are already identified by other works.

2. Only empirical results are provided, there is no theoretical analysis or deep explanation regarding the empirical results, which makes this work less solid.

3. For the experiments, only Deepwalk is compared among all the shallow methods, and only homophilic datasets are used while some heterophilic datasets are missing (e.g. datasets in [3]).

[1] LambdaNet: Probabilistic type inference using graph neural networks. (https://arxiv.org/abs/2005.02161)
[2] Generalized Equivariance and Preferential Labeling for GNN Node Classification. (https://arxiv.org/pdf/2102.11485.pdf)
[3] Beyond Homophily in Graph Neural Networks: Current Limitations and Effective Designs (https://arxiv.org/abs/2006.11468)

**Questions:**

Q1. In difference 1, I personally feel GNN is very flexible to the learning prior. Though one of the most frequently used learning prior would be the transformed node attributes, but it can also take uniform initialization (i.e. treat the input graph as an unattributed graph, then assign uniform initial features on each node). So, it seems to me that, it is unfair to claim GNN is limited to taking the transformed node attributes as prior？

Q2. Only DeepWalk is examined among all the shallow methods. Is it representative enough? Can it outperform all other shallow methods on all datasets? If yes, then why?

---

> ### Author Response · Authors · 2023-11-22
> **Author Response to Reviewer VyuR (1/3)**
>
> We sincerely appreciate the time and efforts you've dedicated to reviewing and providing invaluable feedback to enhance the quality of this paper. We provide a point-to-point reply below for the mentioned concerns and questions.
>
> ---
>
>
> >  **Reviewer**: W1. Novelty is not good. The findings that GNNs face some challenges when we do not have enough attribute information (e.g.[1, 2]) and when we have heterophilic data (e.g.[3]) are already identified by other works.
>
> **Authors**: We agree with the reviewer that (1) existing works such as [1, 2] have pointed out that GNNs face challenges when we lack attribute information, and (2) GNNs also face challenges on heterophilic graphs [3]. However, we note that the main focus of this paper is to perform a comprehensive comparison between GNNs and shallow embedding methods. Especially, to the best of our knowledge, no existing work has comprehensively explored the strengths and weaknesses of GNNs compared with shallow methods with a unified view. Therefore, a comprehensive analysis is still needed to have an in-depth analysis of GNNs. For example, it is valuable to point out that dimensional collapse solely happens in GNNs instead of traditional shallow methods, and such a conclusion has not been revealed by any other existing works.
>
> [1] Wei, J., Goyal, M., Durrett, G., & Dillig, I. (2020). Lambdanet: Probabilistic type inference using graph neural networks. arXiv preprint arXiv:2005.02161.
>
> [2] Sun, Z., Zhang, W., Mou, L., Zhu, Q., Xiong, Y., & Zhang, L. (2022, June). Generalized equivariance and preferential labeling for gnn node classification. In Proceedings of the AAAI Conference on Artificial Intelligence (Vol. 36, No. 8, pp. 8395-8403).
>
> [3] Zhu, J., Yan, Y., Zhao, L., Heimann, M., Akoglu, L., & Koutra, D. (2020). Beyond homophily in graph neural networks: Current limitations and effective designs. Advances in neural information processing systems, 33, 7793-7804.
>
> ---
>
>
> >  **Reviewer**: W2. Only empirical results are provided, there is no theoretical analysis or deep explanation regarding the empirical results, which makes this work less solid.
>
> **Authors**: We agree with the reviewer that the corresponding theoretical analysis regarding the comparative study between GNNs and shallow methods would be great follow-up works. Nevertheless, this paper mainly focuses on an empirical comparison between GNNs and shallow methods. Specifically, a comprehensive analysis is needed to have an in-depth analytical comparison between GNNs and traditional shallow methods. We present an empirical comparison of (1) the levels of dimensional collapse and (2) performances over different levels of homophily between the two types of models. We have obtained consistent observations across different datasets, which makes the discussion self-explanatory and generalizable across different models. As such, we have also provided a practitioner’s guide to help determine which type of model to use. With these contributions, this paper would be particularly interesting to those practitioners.

---

> ### Author Response · Authors · 2023-11-22
> **Author Response to Reviewer VyuR (2/3)**
>
> ---
> >  **Reviewer**: W3. For the experiments, only Deepwalk is compared among all the shallow methods, and only homophilic datasets are used while some heterophilic datasets are missing (e.g. datasets in [3]).
>
> **Authors**: We thank the reviewer for pointing this out. We elaborate on the details corresponding to the two concerns below.
>
> First, the reason why DeepWalk is adopted is that DeepWalk is a representative example of walk-based shallow methods **in its design**. Specifically,  DeepWalk is among the most commonly used shallow graph embedding methods, and a large amount of following works under the umbrella of shallow methods are developed based on DeepWalk, such as [1, 2, 3]. Therefore, **DeepWalk is among the best options we can choose to obtain generalizable analysis**, and adopting more follow-up methods that share similar design with DeepWalk does not change the observation and conclusion.
>
> Second, we note that our analysis **does not depend on whether the adopted datasets are homophilic or not**. As suggested, we perform experiments on the suggested datasets. We select the Squirrel dataset and present the corresponding performances below as a representative example, since the Squirrel dataset has a comparable scale (5,201 nodes) with the datasets adopted in our paper (with a larger scale than most heterophilic graph datasets) and is also highly heterophilic (homophilic ratio 0.22, among the lowest ones). First, we perform experiments to evaluate dimensional collapse. Here utility is measured by node classification accuracy, while the effective dimension ratio (EDR) is measured by the ratio of the value of rank (of representation matrix) to the representation dimensionality. We observe a similar tendency as presented in our paper: the GNN model bears severe dimensional collapse when available attributes become limited, while the shallow method is not influenced since it does not take any node attribute as its input.
>
> |                | 100%Att, Acc | 100%Att, EDR | 1%Att, Acc | 1%Att, EDR | 0.01%Att, Acc | 0.01%Att, EDR |
> | -------------- | ------------ | ------------ | ---------- | ---------- | ------------- | ------------- |
> | GNN            | **38.8%**    | 74.2%        | 26.3%      | 21.9%      | 19.0%         | 1.56%         |
> | Shallow Method | 31.5%        | **99.6%**    | **31.5%**  | **99.6%**  | **31.5%**     | **99.6%**     |
>
> Second, we also perform experiments to study how the performance changes from highly heterophilic nodes to highly homophilic nodes. We first present the study between GNN (w/ neighborhood aggregation) and GNN w/o neighborhood aggregation. We present their cumulative performances in node classification accuracy under different values of the homophilic score below (the same setting as in Fig. 6 in our paper). We observe a similar tendency as presented in our paper: neighborhood aggregation will jeopardize the performances over those highly heterophilic nodes while benefiting highly homophilic nodes.
>
> |                      | 1e-3       | 5e-3       | 1e-2       | 5e-2       | 1e-1       | 5e-1       | 1e0        |
> | -------------------- | ---------- | ---------- | ---------- | ---------- | ---------- | ---------- | ---------- |
> | GNN (w/ Aggregation) | 26.88%     | 26.88%     | 26.88%     | 26.73%     | 25.44%     | **36.64%** | **38.75%** |
> | GNN w/o Aggregation  | **30.10%** | **30.10%** | **30.10%** | **30.69%** | **26.48%** | 32.20%     | 33.17%     |
>
> We also perform experiments to compare the shallow method w/ neighborhood aggregation vs. the shallow method w/o neighborhood aggregation, and the observations remain consistent.
>
> |                           | 1e-3       | 5e-3       | 1e-2       | 5e-2       | 1e-1       | 5e-1       | 1e0        |
> | ------------------------- | ---------- | ---------- | ---------- | ---------- | ---------- | ---------- | ---------- |
> | Shallow w/ Aggregation    | 21.86%     | 21.86%     | 21.86%     | 23.23%     | 26.35%     | **30.36%** | **31.54%** |
> | Shallow (w/o Aggregation) | **25.14%** | **25.14%** | **25.14%** | **26.26%** | **26.71%** | 29.95%     | 31.44%     |
>
> In conclusion, we argue that, first, DeepWalk is **representative enough** to obtain generalizable experimental results, and adopting more follow-up methods does not change the conclusions; Second, we also have **similar observations** on heterophilic datasets from both studied perspectives, and our analysis does not depend on whether the adopted datasets are homophilic or not.
>
> We thank the reviewer again for bringing this up. We have included the experimental results and the corresponding discussion in Appendix C.7 and C.8.
>
> [1] Grover et al. node2vec: Scalable feature learning for networks. In SIGKDD 2016.
>
> [2] Perozzi et al. Walklets: Multiscale graph embeddings for interpretable network classification. arXiv preprint arXiv:1605.02115, 043238-23.
>
> [3] Huang et al. Hyper2vec: Biased random walk for hyper-network embedding. In DASFAA 2019.

---

> ### Author Response · Authors · 2023-11-22
> **Author Response to Reviewer VyuR (3/3)**
>
> ---
>
>
> >  **Reviewer**: Q1. In difference 1, I personally feel GNN is very flexible to the learning prior. Though one of the most frequently used learning prior would be the transformed node attributes, but it can also take uniform initialization (i.e. treat the input graph as an unattributed graph, then assign uniform initial features on each node). So, it seems to me that, it is unfair to claim GNN is limited to taking the transformed node attributes as prior？
>
> **Authors**: We agree with the reviewer that GNNs do not necessarily take node attributes as prior. We have improved the expression in our paper accordingly. However, we also note that taking both graph topology and node attributes as the input of GNNs is the **most widely studied scenario**. In fact, being able to utilize node attributes is considered an advantage of GNNs in most cases compared with most traditional methods that only take graph topology as input. If node features are already available, avoiding using them usually leads to suboptimal performances.
>
> ---
>
> >  **Reviewer**: Q2. Only DeepWalk is examined among all the shallow methods. Is it representative enough? Can it outperform all other shallow methods on all datasets? If yes, then why?
>
> **Authors**: We thank the reviewer for pointing this out. The reason why DeepWalk is adopted is that DeepWalk is a representative example of shallow methods **in its design**. Specifically,  DeepWalk is among the most commonly used shallow graph embedding methods, and a large amount of following works under the umbrella of shallow methods are developed based on DeepWalk, such as [1, 2, 3]. Therefore, **DeepWalk is among the best options we can choose to obtain generalizable analysis**, and adopting more follow-up methods that share a similar design with DeepWalk does not change the observation and conclusion.
>
> In addition, we note that a model with representative design does not necessarily have SOTA performances. For example, we do not examine SOTA shallow methods with a series of unique designs in this paper (e.g., [4]), since the analysis performed on these models may not be generalizable to other shallow methods.
>
> We thank the reviewer again for bringing this up so we can make our evaluation even more comprehensive. We have included the experimental results and the corresponding discussion in Appendix C.7.
>
> [1] Grover, A., & Leskovec, J. (2016, August). node2vec: Scalable feature learning for networks. In Proceedings of the 22nd ACM SIGKDD international conference on Knowledge discovery and data mining (pp. 855-864).
>
> [2] Perozzi, B., Kulkarni, V., & Skiena, S. (2016). Walklets: Multiscale graph embeddings for interpretable network classification. arXiv preprint arXiv:1605.02115, 043238-23.
>
> [3] Huang, J., Chen, C., Ye, F., Wu, J., Zheng, Z., & Ling, G. (2019). Hyper2vec: Biased random walk for hyper-network embedding. In Database Systems for Advanced Applications: DASFAA 2019 International Workshops: BDMS, BDQM, and GDMA, Chiang Mai, Thailand, April 22–25, 2019, Proceedings 24 (pp. 273-277). Springer International Publishing.
>
> [4] Postăvaru, Ş., Tsitsulin, A., de Almeida, F. M. G., Tian, Y., Lattanzi, S., & Perozzi, B. (2020).  InstantEmbedding: Efficient local node representations. arXiv preprint arXiv:2010.06992.

---

> ### Author Response · Authors · 2023-11-22
> **A kind reminder**
>
> Dear Reviewer VyuR,
>
> We would like to express our sincere gratitude to you for reviewing our paper and providing valuable feedback. We believe that we have responded to and addressed all your concerns with our revisions — in light of this, we hope you consider raising your score.
>
> Notably, given that we are approaching the deadline for the rebuttal phase, we hope we can have the discussion soon. Thanks！
>
> Best,
> All authors

---

### Author Response · Authors · 2023-11-22
**Official Comment by Authors**

We thank all the reviewers for their constructive feedback and thoughtful comments on our work.

We have provided a point-to-point reply to each of the reviewers individually to address the mentioned concerns and questions. We have also made updates to our paper according to the great suggestions from the reviewers in blue font color. We look forward to hearing feedback from the reviewers and engaging in further discussion.

---

### Meta-Review · Area_Chair_CyBL · 2023-12-10

**Metareview:**

This submission presents a comparative study of Graph Neural Networks (GNNs) and shallow embedding methods in the context of node representation learning. The authors introduce SEESAW, a framework designed to systematically evaluate the performance of these two approaches across various scenarios. The paper's primary focus is on identifying conditions under which GNNs may not outperform shallow embedding methods, particularly in attribute-poor scenarios and highly heterophilic networks.

Reviewer VyuR raises concerns about the novelty of the work, noting that some findings are already established in prior research. The reviewer also points out the lack of theoretical analysis and the limited scope of empirical comparisons, specifically questioning the choice of only comparing DeepWalk among shallow methods and the absence of heterophilic datasets in the experiments. Reviewer 4Dj2 acknowledges the importance of the topic and the clarity of the paper but echoes the concern regarding the novelty of the findings. The reviewer also questions the practical applicability of the guidelines provided for practitioners, noting a lack of specificity in the recommendations. Reviewer 4Gbd appreciates the paper's clarity and validation on various datasets but notes that the findings might not be surprising or particularly novel, given existing literature. The reviewer also questions the real-world applicability of the scenarios where input features are limited. Reviewer ySnn commends the detailed analysis and the insights provided into the strengths and weaknesses of GNNs and shallow methods. However, the reviewer also notes the lack of novelty and suggests that the paper could be improved by discussing works that combine the advantages of both approaches.

The authors provided detailed responses to the reviewers' concerns, including additional experiments. However, the responses seem to have only partially addressed the concerns. Reviewers 4Dj2 and ySnn maintained their scores, citing the limited novelty and practical applicability of the findings. Reviewer 4Gbd slightly raised their score, acknowledging the authors' efforts in addressing some concerns but still questioning the broad impact of the findings.

The paper tackles an important question in the field of graph representation learning and is well-structured and clearly written. The experimental setup represents serious efforts, and the provision of code is commendable. However, the concerns raised by the reviewers about the novelty of the findings and the practical applicability of the recommendations are significant. The paper seems to reiterate known challenges with GNNs in certain scenarios without providing substantial new insights or theoretical underpinnings to advance the understanding of these issues. While the paper contributes to the ongoing discussion about the effectiveness of GNNs compared to shallow embedding methods, its contributions appear incremental and somewhat limited in scope. The lack of novelty and theoretical depth, coupled with concerns about the practical applicability of its recommendations, suggests that the paper may not meet the high standards of ICLR in its current form.

**Justification For Why Not Higher Score:**

The concerns raised by the reviewers about the novelty of the findings and the practical applicability of the recommendations are significant. The paper seems to reiterate known challenges with GNNs in certain scenarios without providing substantial new insights or theoretical underpinnings to advance the understanding of these issues. While the paper contributes to the ongoing discussion about the effectiveness of GNNs compared to shallow embedding methods, its contributions appear incremental and somewhat limited in scope.

**Justification For Why Not Lower Score:**

N/A

---

### Decision · Program_Chairs · 2024-01-16

Reject